# Escaping the SpuriVerse: Can Large Vision-Language Models Generalize Beyond Seen Spurious Correlations?

**Yiwei Yang**[1,*], **Chung Peng Lee**[4,*], **Shangbin Feng**[1], **Dora Zhao**[2], **Bingbing Wen**[1],
**Anthony Z. Liu**[3], **Yulia Tsvetkov**[1], **Bill Howe**[1]

[1]University of Washington   [2]Stanford University   [3]University of Michigan
[4]Princeton University

yanyiwei@uw.edu, cl6486@princeton.edu, shangbin@cs.washington.edu,
dorothyz@stanford.edu, bingbw@uw.edu, anthliu@umich.edu,
yuliats@cs.washington.edu, billhowe@uw.edu

## Abstract

Spurious correlations occur when models rely on non-essential features that coincidentally co-vary with target labels, leading to incorrect reasoning under distribution shift. We consider spurious correlations in Large Vision Language Models (LVLMs) pretrained on extensive and diverse datasets without explicit task supervision. We develop a benchmark by sourcing GPT-4o errors on real-world visual-question-answering (VQA) benchmarks, then curating a subset through LVLM-human annotation and synthetic counterfactual evaluation to identify errors caused by spurious correlations. This process yields SpuriVerse, a novel benchmark comprised of 124 distinct types of spurious correlations extracted from real-world datasets, each containing 1 realistic and 10 synthetic VQA samples for a total of 1364 multiple choice questions. We evaluate 15 open and closed-source LVLMs on SpuriVerse, finding that even state-of-the-art closed-source models struggle significantly, achieving at best only 35.0% accuracy. Fine-tuning on synthetic examples that emphasize the spurious correlation improves performance to 78.4%, suggesting that training on diverse spurious patterns generalizes to unseen situations: models appear to learn to avoid "shortcuts" and attend to the overall image context.

## 1 Introduction

Real-world data frequently contains spurious correlations —- patterns predictive of the target during training but irrelevant to the true label [Leino et al., 2018]. For example, in ImageNet, butterflies often appear with flowers [Singla and Feizi, 2021], while the Waterbirds dataset was constructed to have waterbird images commonly appear on water backgrounds [Sagawa et al., 2019]. Models trained on such data may learn to rely on these *spurious features* (backgrounds, co-occurring objects, texture, etc.), leading to errors when the correlations no longer hold. An example of such errors is a landbird incorrectly classified as a waterbird simply due to its water background [Sagawa et al., 2019]. This phenomenon is especially problematic in critical domains such as medicine, where models trained to detect pneumonia might rely on unrelated features (e.g., metal tokens) rather than genuine disease indicators [Zech et al., 2018].

---

[*]Equal contribution.

39th Conference on Neural Information Processing Systems (NeurIPS 2025) Track on Datasets and Benchmarks.

In traditional supervised training where a model is finetuned for a downstream application, spurious correlation arises from the association between spurious features and *target labels* captured by the model through standard loss minimization [Sagawa et al., 2019, Kirichenko et al., 2022]. In the regime of pre-training Large Vision-Language Models (LVLMs), not only are the training data more diverse and extensive, but the objective is no longer predicting a task-specific target label. In this generalized regime, we hypothesize that correlations between spurious features and target concepts can still persist and compromise LVLMs' performance. However, modern LVLMs' evaluation suites do not examine whether zero-shot LVLMs rely on spurious features to make incorrect predictions, where fine-tuning for a particular target label is not available. In this work, we **provide a benchmark to evaluate an LVLM's susceptibility to spurious correlations in generalized settings.** In particular, we are interested in spurious correlations that appear outside of contrived tasks where the correlation is a result of non-representative training data such as `Gender-to-BlondHair` in CelebA [Liu et al., 2015] and `Background-to-BirdType` in WaterBirds [Sagawa et al., 2019]. Instead of contrived spurious correlations, we focus on situations where a model may overrely on a *dominant* correlation that is not necessarily irrelevant but is not generally reliable (Figure 1).

We address two important challenges: (1) Existing benchmarks often study spurious correlations by collecting a training set with an imbalanced distribution across spurious features and target labels [Sagawa et al., 2019, Lynch et al., 2023, Joshi et al., 2023, Li et al., 2023b, Arjovsky et al., 2019]. These benchmarks are particularly useful for investigating whether a model learns spurious correlations when trained on a skewed distribution. However, when evaluating pre-trained LVLMs, curating an imbalanced dataset is ineffective due to distribution shifts. (2) The presence of spurious correlations often relies on annotations of *spurious features* and *target label*. For example, in CelebA [Liu et al., 2015], the comprehensive list of annotated face attributes allows one to recognize a strong correlation between *gender* and *hair color* in the training set. Observing that the trained model achieves low accuracy on non-blond males then confirms that the spurious correlation is captured by the model. However, this approach only applies to a specific target label identified beforehand. While one may consider annotating an extensive list of potential spurious features, and examining correlation among all possible pairs, it is still impractical to annotate them on LVLM's web-scale pre-training data.

Instead of guessing the candidates for *spurious features*, we choose a bottom-up approach, described in Figure 2: we (1) identify errors of a strong LVLM on real-world visual-question-answering (VQA) benchmarks; (2) derive a subset of errors attributable to spurious correlations via human-LVLM collaboration; then (3) validate these derived samples by evaluating multiple LVLMs on synthetic counterfactual examples with and without the spurious features. A significant accuracy drop when spurious feature(s) are present indicates the model's reliance on spurious correlation. This process yields SpuriVerse, a novel multimodal benchmark comprised of 124 distinct spurious correlations extracted from real-world datasets [Li et al., 2024a, Schwenk et al., 2022, Li et al., 2023a, 2024b], each containing 11 VQA samples—1 from the original dataset, referred to as the *anchor* and 10 synthetic examples, generated as part of the validation process, referred to as the *spurious group*— for a total of 1364 multiple choice questions involving visual understanding. SpuriVerse affords evaluation of the susceptibility of LVLMs to diverse spurious correlations in a generalized setting. Moreover, this diversity enables investigation of **whether models can learn to ignore spurious correlations as a meta-skill to generalize to unseen situations**.

We evaluate on 15 LVLMs, finding that even state-of-the-art closed-source models such as Qwen-VL-Max struggle significantly, achieving only 35.0% accuracy. No prompt-based methods such as Chain-of-Thought [Wei et al., 2022] effectively mitigates the vulnerability. However, fine-tuning on a subset of SpuriVerse's *anchor* and evaluating on the held-out set of *anchor*, improves accuracy from 35.20% to 45.60% on previously unseen (not fine-tuned) spurious correlations for Qwen-2.5-VL-7b-Instruct. Furthermore, we boost the performance to 78.40% by fine-tuning the same model on the *spurious group* counterparts of the original training subset, where the held-out set is still previously unseen spurious correlations. These results suggest that models can learn a general meta-skill to avoid being distracted by a dominant correlation and pay more attention to the overall scene. However, we also observe a trade-off in overall performance (14% on Qwen-2.5-VL-7b-Instruct), suggesting that models rely on these "shortcuts" to deliver their performance.

Our contributions are three-fold: (1) introducing a first spurious correlation benchmark centering diverse, natural, general Q&A settings suitable for frontier LVLMs, (2) demonstrating that fine-tuning models on a diverse set of images emphasizing a spurious correlation improves accuracy on examples

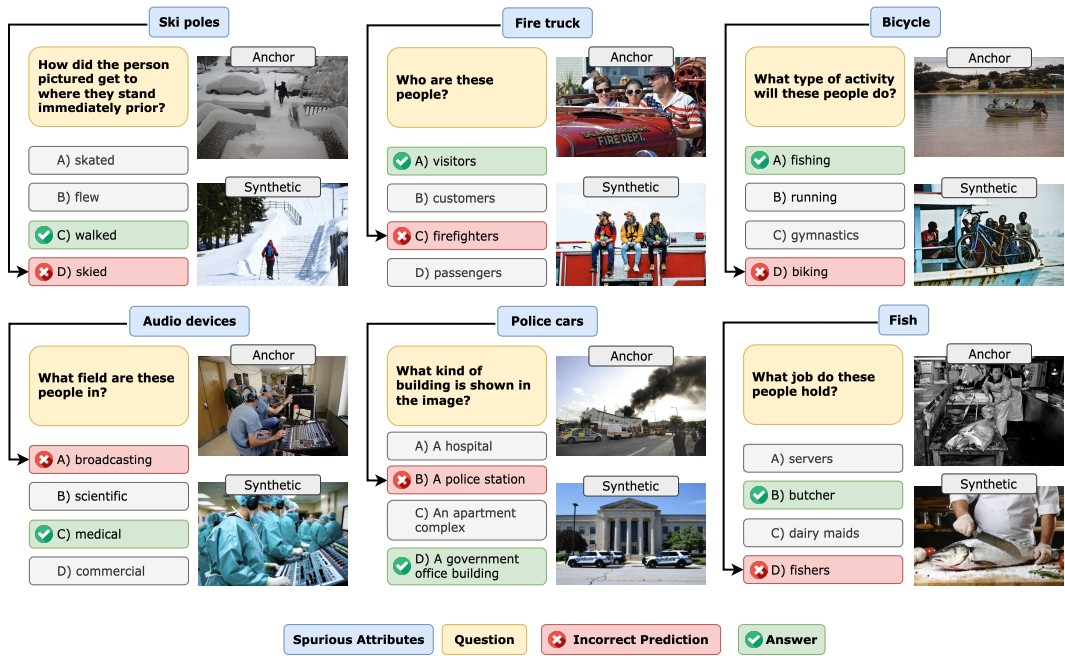

Figure 1: **SpuriVerse** consists of 124 distinct spurious correlations, each of which is comprised of 1 original example and 10 synthetic images. Both original and synthetic images share the same multiple-choice question. Each image contains a feature that is spuriously correlated with one of the choices, causing the model to make an error.

with *unseen* spurious features, and (3) revealing a trade-off between performance on samples with and without spurious features. These findings suggest a relationship between spurious features and visual understanding: when models are penalized for taking shortcuts, performance suffers.

## 2 SpuriVerse

### 2.1 Motivation

In the task-oriented setting, spurious correlations arise from non-representative correlations between features and target labels in the training data. While spurious correlations surely exist in large-scale and diverse pre-training data, it is impractical to identify a representative set of specific correlations and use them for mitigation. We instead adopt a data-driven approach, using existing LVLM benchmarks that assess a variety of model capabilities, then identify which errors are attributable to spurious correlations.

### 2.2 Curation Pipeline

**Step 1: Select a challenging set.** We start with the error set of a strong model, GPT-4o, on a collection of commonly used multi-modal multiple-choice question-answering benchmarks, including SEEDBench [Li et al., 2023a], SEEDBench2 [Li et al., 2024b], NaturalBench [Li et al., 2024a], and AOKVQA [Schwenk et al., 2022]. This process produces 11225 samples in the error set from 55911 total samples.

**Step 2: Two-stage verification.** Given these challenging samples, we first prompt GPT-4o to examine whether the error can be attributed to spurious correlation and, if so, to propose the candidate spurious features. This process results in 1717 samples being *VLM Accepted*. Human annotators then review each sample to 1) validate that the error can be attributed to spurious correlation and 2) refine the proposed spurious features if necessary. This process produces 194 samples being *Human Accepted*.

**Step 3: Generate counterfactual scene descriptions.** To verify that the error is attributable to the identified spurious attribute, we generate counterfactual images based on the question and answer

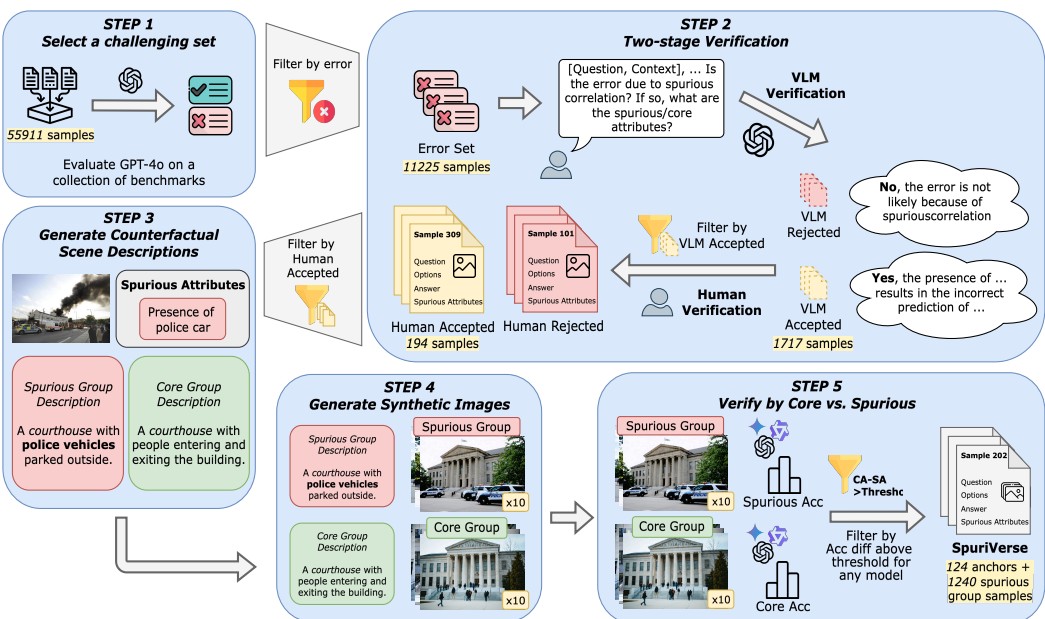

Figure 2: **Curating SpuriVerse** consists of 5 steps: (1) Derive errors from GPT-4o on a set of multi-modal multiple choice question benchmarks. (2) Two-stage Verification: (a) Prompt GPT-4o to identify errors attributable to spurious correlation and report the candidate spurious feature, and (b) Human verify the attribution and refine the spurious features if necessary. (3) Prompt GPT-4o to generate two scene descriptions based on the correct answer, one with and one without the spurious feature. (4) Generate 10 synthetic images for each description in pairs using Stable Diffusion, forming spurious and core groups for each sample. (5) Evaluate multiple LVLMs on both groups, and select those samples for which the accuracy difference is at least 30% for any model.

that does *not* include the spurious feature. For example, a person holding an umbrella may mislead the model into incorrectly guessing the weather is rainy despite visible sunlight; we seek images for which the answer is "sunny" both with and without an umbrella.

Specifically, for each image-question-answer triple $(i, q, a)$, we prompt an LLM to generate a *core group description* (CGD) and a *spurious group description* (SGD). For both the CGD and the SGD, the prompt includes the question $q$ and the answer $a$, but for the SGD, the model is instructed explicitly to include the spurious attribute. The specific prompt can be found in the Appendix. The result is that for each sample $(i, q, a)$ there are two associated scene descriptions $d_i^{\text{spurious}}$ and $d_i^{\text{core}}$.

**Step 4: Generate Synthetic Images.** We use the pairs of scene descriptions from Step 3 to generate a set of 10 core images (the *core group*) and 10 spurious images (the *spurious group*) using Stable Diffusion [Podell et al., 2023]. We employ human verification to ensure that the generated images are faithful to the scene descriptions with the appropriate spurious and core features, where *core features* means features associated with the correct answer. We manually edit the scene descriptions to improve their faithfulness if applicable. At this point, each sample $s_i$ has two sets of images, $G_i^{\text{spurious}}$ and $G_i^{\text{core}}$, where the size of each group is 10.

**Step 5: Verify by Core vs. Spurious.** Since the spurious feature was removed, we expect that performance on the counterfactual core group should be higher than that of the spurious group. To validate this assumption, we measure accuracy on both groups and only retain those samples that exhibit at least a 30% difference in accuracy in at least one of GPT-4o, Gemini 2.0 Flash, or Qwen-VL-Max.

Concretely, let $Acc(f, G, s)$ be a function that represents the accuracy of model $f$ on a group of images $G$ associated with sample $s$. The selected samples for a particular model $f$ are $\mathcal{S}_{\text{anchor}, f}$ : $\{s_i | Acc(f, G_i^{\text{core}}, s_i) - Acc(f, G_i^{\text{spurious}}, s_i) \geq \epsilon\}$. Aggregating on the set $\mathcal{F}$ yields the final anchor set

$$\mathcal{S}_{\text{anchor}} = \bigcup_{f \in \mathcal{F}} \mathcal{S}_{\text{anchor},f} = \left\{ s_i \; \middle| \; \max_{f \in \mathcal{F}} \left[ Acc(f, G_i^{\text{core}}, s_i) - Acc(f, G_i^{\text{spurious}}, s_i) \right] \geq \epsilon \right\}$$

## 2.3 Categories of spurious correlations

As discussed in prior work [Geirhos et al., 2020], spurious correlations can be grouped into several high-level categories. To systematically understand the types of spurious correlations commonly found in real-world datasets, we presented all 124 identified correlations to GPT-4.5, requesting that it categorize each one. We manually reviewed each category label for correctness. Table 1 shows the categories, and Figure 3a shows the proportion of each category of spurious correlations in SpuriVerse.

| Category | Description | Example |
|---|---|---|
| Object co-occurrences | Model incorrectly predicts based on commonly associated objects appearing together. | Misclassifying a person as skiing when they're walking with ski poles. |
| Contextual cues | Model mispredicts based on background context rather than the main subject itself. | Misclassifying a person as wet because they're near a lake. |
| Visual predominance | Model misclassifies due to focusing on the most visually dominant object in the scene. | Misclassifying someone sitting in the corner due to a figure jumping centrally. |
| Physical properties | Model confuses items based on their texture, material, or other physical attributes. | Misclassifying a scarf as a stuffed toy. |
| Visual resemblance | Model mistakes an object for another because they look visually similar. | Misclassifying a mannequin as a real person. |
| Spatial relationships | Model misinterprets actions or interactions based on object positions or orientations. | Misclassifying someone as kicking another due to proximity of their legs. |

Table 1: **Categories of spurious correlations** identified in SpuriVerse.

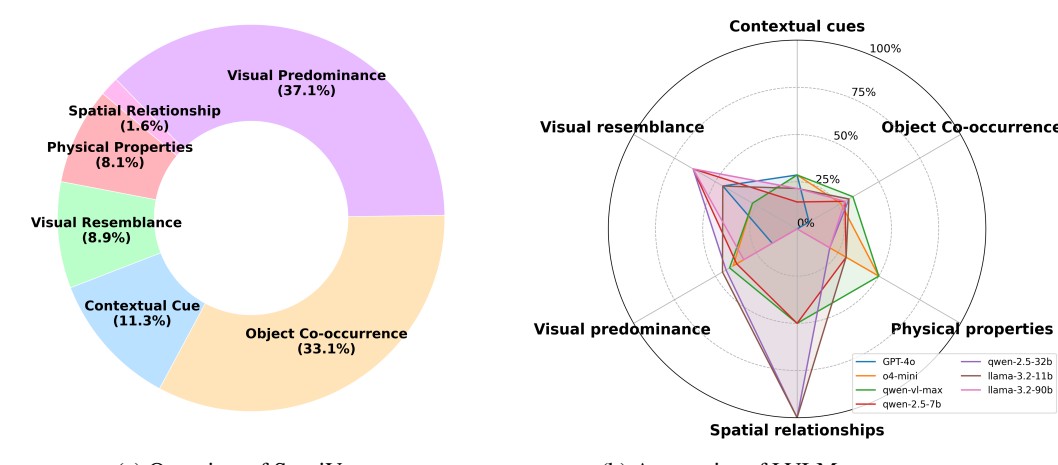

(a) Overview of SpuriVerse

(b) Accuracies of LVLMs per category

Figure 3: **Categories of spurious correlations**. Visual predominance and Object co-occurrence make up 37.1%, 33.1% of SpuriVerse, respectively. Only 2 spurious correlations fall under Spatial relationships. LVLMs tend to do better on Visual resemblance and Spatial relationships, achieving more than 50% accuracy, and achieve less than 50% accuracy on all other categories.

## 3 Experiments

In this section, we first present a comprehensive evaluation of 15 recent LVLMs on SpuriVerse. We then show that prompting methods are ineffective in correcting spurious correlations. However, we

demonstrate that simply finetuning on a diverse set of spurious correlations helps generalize to unseen spurious correlations. Lastly, by increasing the proportion of spurious correlation samples in the finetuning set, we observe an increase in the model's accuracy on held-out spurious correlations, and a trade-off in the accuracy on the non-spurious samples.

## 3.1 Evaluation of SOTA LVLMs on SpuriVerse

We evaluate SpuriVerse on 15 recent LVLMs, including GPT-4o, GPT-4o-mini, o4-mini, o3, Gemini 2.0 Flash, Gemini 1.5 Pro, Claude 3.7 Sonnet, Qwen-VL-Max, Qwen-VL-Pro, Qwen2.5-vl-7b-instruct, Qwen2.5-vl-32b-instruct, Llama-3.2-11B-vision-instruct, Llama-3.2-90B-vision-instruct, LLaVA-v1.6 7b and LLaVA-v1.5. We present results on the anchor set (124 samples), their corresponding spurious groups (124 x 10 samples), and non-spurious samples drawn randomly from the source benchmarks (1240, same benchmark distribution as the anchor set).

Table 2 shows that all LVLMs perform worse than random guess on the anchor set. Two open-source models, Llama-3.2-11B-vision-instruct and Qwen2.5-vl-32b-instruct achieve the highest accuracy of 35.0%, lower than random guess by 5.7%. The LVLMs perform slightly better on the spurious group, with o4-mini achieving the highest accuracy of 50.45%, outperforming random guess by 9.75%. The increase in accuracy is expected because the spurious group is synthesized based on attributes extracted from the anchors, which inherently leads to a decrease in complexity. All LVLMs perform significantly better on non-spurious samples. The accuracy gap between the anchor set and non-spurious samples ranges from 39.65% to 64.92%. The accuracy gap between the spurious groups and non-spurious samples ranges from 33.24% to 51.63%. These results demonstrate that a) the spurious samples are challenging for all models, and that b) the difficulty is attributable to the presence of the spurious attribute.

In addition, Figure 3b shows the accuracies of select models per category. Overall, LVLMs tend to perform better on Spatial relationships and Visual resemblance. The especially high accuracy of Llama-3.2-11b and Llama-3.2-90b on Spatial relationships is likely because only 2 spurious correlations fall under this category. For the other four categories (Contextural cues, Object co-occurrence, Physical properties, Visual predominance), all models obtain less than 50% accuracy.

| Model | Anchor (%) | Spurious (%) | Non-spurious (%) | $\Delta_{\text{NS-A}}$(%) | $\Delta_{\text{NS-S}}$(%) |
|---|---|---|---|---|---|
| **Open-Source Models** | | | | | |
| llama-3.2-11b | $25.32_{0.82}$ | $29.89_{0.14}$ | $73.21_{0.25}$ | 47.89 | **43.32** |
| llama-3.2-90b | $31.45_{1.25}$ | $36.74_{0.41}$ | $79.35_{0.15}$ | **47.90** | 42.61 |
| qwen-2.5-7b | $\mathbf{35.00}_{0.65}$ | $41.60_{0.52}$ | $77.87_{0.24}$ | 42.87 | 36.27 |
| qwen-2.5-32b | $33.06_{0.72}$ | $\mathbf{44.97}_{0.33}$ | $\mathbf{80.74}_{0.49}$ | 47.68 | 35.77 |
| llava-1.5 | $28.71_{1.09}$ | $28.18_{0.46}$ | $68.35_{0.73}$ | 39.65 | 40.18 |
| llava-1.6 | $29.68_{2.99}$ | $28.61_{1.13}$ | $71.32_{0.75}$ | 41.65 | 42.71 |
| **Closed-Source Models** | | | | | |
| qwen-vl-max | $27.10_{0.97}$ | $43.32_{0.45}$ | $78.90_{0.08}$ | 51.81 | 35.58 |
| qwen-vl-plus | $28.06_{0.94}$ | $35.92_{0.43}$ | $71.50_{0.06}$ | 43.44 | 35.58 |
| gpt-4o | $17.26_{1.81}$ | $47.21_{1.24}$ | $82.18_{0.71}$ | **64.92** | 34.97 |
| gpt-4o-mini | $18.87_{1.66}$ | $23.95_{1.00}$ | $75.58_{0.48}$ | 56.71 | **51.63** |
| gemini-1.5-pro | $21.77_{0.00}$ | $40.19_{0.06}$ | $78.02_{0.06}$ | 56.24 | 37.82 |
| gemini-2.0-flash | $25.32_{1.31}$ | $35.27_{0.63}$ | $81.89_{0.37}$ | 56.56 | 46.61 |
| claude-3.7-sonnet | $25.65_{1.07}$ | $41.21_{0.46}$ | $76.69_{0.37}$ | 51.05 | 35.48 |
| o4-mini | $\mathbf{33.87}_{1.35}$ | $\mathbf{50.45}_{0.72}$ | $83.69_{0.56}$ | 49.82 | 33.24 |
| o3 | $31.94_{2.08}$ | $50.34_{0.38}$ | $\mathbf{85.05}_{0.78}$ | 53.11 | 34.71 |
| Random | 40.70 | 40.70 | 40.70 | 0.00 | 0.00 |

Table 2: **Performance on SpuriVerse**. We report the performance of 15 recent LVLMs on SpuriVerse. All LVLMs perform worse than random guess on the anchor set. Even the best model achieves the highest accuracy of 35.0%, lagging behind random guess by 5.7%. The LVLMs perform poorly on spurious group as well, with the best model o4-mini achieving 50.45%, outperforming random guess by merely 9.75%. As a point of comparison, the LVLMs all perform significantly better on non-spurious samples. ($\Delta_{\text{NS-A}}$: Non-spurious - Anchor, $\Delta_{\text{NS-S}}$: Non-spurious - Spurious).

## 3.2 Effectiveness of prompting methods on spurious correlation

We explore two prompting methods for correcting spurious correlations. One method we evaluated is Chain-of-Thought [Wei et al., 2022], which asks the model to first give its reasoning step by step, then give a final answer. Further, we designed a prompt strategy specifically for correcting spurious correlations, which we term *spurious-aware*. Spurious aware instructs the model to be aware that there may be spurious features in the image, asks them to first describe the potential spurious features, then make a prediction without relying on the spurious features. We evaluated on GPT-4o, Llama-3.2-11B-vision-instruct, and Qwen2.5-vl-7b-instruct on the anchor set, spurious groups, and non-spurious samples, same as the main evaluation.

As shown in Table 3, Chain-of-Thought shows insignificant improvements on both anchor set and spurious groups for all three models, while Spurious Aware shows moderate improvements. However, even the best combination of Qwen2.5-vl-7b-instruct and Spurious Aware achieves only 51.61% on the anchor set, merely 10.91% above random guess. These results suggest that prompting alone is unlikely to provide a general solution to the problem.

| Model | Prompt Strategy | Anchor (%) | Spurious (%) | Non-spurious (%) |
|---|---|---|---|---|
| GPT-4o | Direct Prompting | 16.13 | 42.90 | **80.48** |
| | Chain of Thought | 25.00 | 48.39 | 77.74 |
| | Spurious Aware | **41.94** | **58.06** | 78.55 |
| Llama-3.2-11b | Direct Prompting | 35.48 | 29.84 | 69.60 |
| | Chain of Thought | 33.06 | 38.23 | **70.48** |
| | Spurious Aware | **50.00** | **52.50** | 69.35 |
| Qwen-2.5-7b | Direct Prompting | 37.90 | 41.77 | 70.24 |
| | Chain of Thought | 37.90 | 48.79 | **74.68** |
| | Spurious Aware | **51.61** | **56.94** | 72.18 |

Table 3: **Effectiveness of prompting strategies.** Chain of Thought shows insignificant improvement, while Spurious Aware moderately improves over Direct Prompting.

## 3.3 Finetuning generalizes to unseen spurious correlation

Can a diverse set of spurious correlations be used to help the model generalize to unseen spurious correlations? We explore this question by finetuning on subsets of SpuriVerse and test on the held-out spurious correlations. We divided both the anchor set and the spurious groups into train/val/test sets according to the ratio of 70/10/20. We finetuned Llama-3.2-11B-vision-instruct and Qwen2.5-vl-7b-instruct on the train and val sets of anchors and spurious groups, respectively, and evaluated on the test sets. As a baseline, we also considered the "non-spurious set", which was previously used to evaluate LVLMs. Similarly, the non-spurious set is further split according to the ratio of 70/10/20.

Table 4 shows that finetuning on the anchor set improves performance on held-out anchors and spurious samples for both models. For example, finetuning Llama-3.2-11B-vision-instruct increases accuracy from 41.60% to 59.20% on anchors, and from 39.20% to 60.48% on spurious samples. On the other hand, finetuning on non-spurious samples shows only slight improvement for Llama-3.2-11B-vision-instruct and slight performance decline for Qwen2.5-vl-7b-instruct. In particular, Llama-3.2-11B-vision-instruct's accuracy increases to 48.80% on anchors, and 51.36% on spurious samples, while Qwen2.5-vl-7b-instruct's accuracy decreases from 35.20% to 34.40% on anchors, and from 41.92% to 38.24% on spurious samples.

While finetuning on the original anchor set improves performance, finetuning on synthetically generated spurious samples produces substantially greater improvements. Llama-3.2-11B-vision-instruct improves from 41.60% to 80.00% on anchors and from 39.20% to 79.12% on spurious samples. This suggests that synthetic examples, which allow us to increase the training size, are effective for generalizing to unseen spurious correlations. However, we observe that the dramatic increase in the model's accuracy on spurious correlations comes at the cost of accuracy on non-spurious samples. In particular, Llama-3.2-11B-vision-instruct's accuracy drops from 73.44% to

| Finetune Setup | Anchor | | Spurious | | Non-spurious | |
|---|---|---|---|---|---|---|
| | **llama** | **qwen** | **llama** | **qwen** | **llama** | **qwen** |
| Anchor | $59.20_{7.76}$ | $45.60_{6.97}$ | $60.48_{3.02}$ | $45.12_{6.79}$ | $66.48_{3.69}$ | $\mathbf{81.36}_{2.91}$ |
| Spurious | $\mathbf{80.00}_{9.12}$ | $\mathbf{78.40}_{3.20}$ | $\mathbf{79.12}_{5.52}$ | $\mathbf{75.60}_{6.20}$ | $56.80_{2.10}$ | $67.20_{3.65}$ |
| Non-spurious | $48.80_{6.40}$ | $34.40_{4.08}$ | $51.36_{2.70}$ | $38.24_{6.95}$ | $71.20_{2.16}$ | $79.84_{2.38}$ |
| Mixed | $71.20_{3.92}$ | $64.80_{6.40}$ | $68.88_{7.75}$ | $64.64_{9.54}$ | $64.72_{1.09}$ | $74.48_{2.56}$ |
| No finetuning | $41.60_{6.50}$ | $35.20_{2.99}$ | $39.20_{3.29}$ | $41.92_{6.00}$ | $\mathbf{73.44}_{3.36}$ | $81.20_{1.84}$ |

Table 4: **Finetuning generalizes** to unseen spurious correlations. When models are finetuned on the anchor set, their performance improves on both held-out anchors and spurious samples. Finetuning directly on spurious samples further boosts accuracy for both models. Compared to finetuning solely on spurious samples, finetuning on the Mixed set significantly enhances performance on non-spurious samples, albeit with some trade-off in accuracy on the anchor set and spurious groups.

56.80% when finetuned on spurious samples. Its accuracy drops to 66.48% when finetuned on anchors.

To balance the trade-off between accuracies on spurious and non-spurious samples, we consider finetuning on a "Mixed set", which is simply a concatenation of Spurious and Non-spurious samples. In comparison to finetuning on spurious samples only, finetuning on the Mixed set greatly improves models' accuracies on non-spurious samples, while trading-off some performance on the anchor set and spurious samples. Specifically, finetuning on mixed improves Llama-3.2-11B-vision-instruct's accuracy from 41.6% to 71.2% on anchors, and 39.2% to 68.88% on spurious groups, lagging behind finetuning on spurious groups by 8.8% on anchors, and 10.24% on spurious groups, while improving from 56.80% to 64.72% on non-spurious samples.

## 3.4 Trade-off between accuracies on spurious and non-spurious samples

We investigate the trade-off between accuracies on spurious and non-spurious samples further by controlling the proportion of spurious samples in the finetuning set. We focus on the spurious group as it demonstrates significant improvement as well as a significant trade-off in table 4. We split the spurious group into train/test with a ratio of 80/20. Then, for the train set, we keep a fraction of the set, and replace the others with random samples from the source benchmarks. We vary the fraction from {0%, 20%, 40%, 60%, 80%, 100%}. Then, the train set is further split into 87.5/12.5. So that the final train/val/test follows the ratio of 70/10/20, while keeping the distribution of train and val set the same. All splits and removals are performed on the type of spurious correlation rather than individual samples to ensure we are measuring generalization to unseen cases.

Figure 4 shows that, as the number of spurious correlations increases, both Llama-3.2-11B-vision-instruct and Qwen2.5-vl-7b-instruct's accuracies increase on both the anchor set and the spurious groups, suggesting that diversity of spurious correlation can indeed improve generalization. Further, both models' accuracy decreases on the non-spurious samples, suggesting that the models may rely on "shortcuts" to achieve high performance.

## 4 Related Work

**Benchmarks for Spurious Correlation** Benchmarks for studying spurious correlation in a classification setting are common [Arjovsky et al., 2019, Joshi et al., 2023, Li et al., 2023b, Koh et al., 2021, Lynch et al., 2023, Ye et al., 2024, Zech et al., 2018]. Two popular vision benchmarks are Waterbirds [Sagawa et al., 2019] and CelebA [Liu et al., 2015]. Waterbirds is a dataset of a water/land bird superimposed on either a water or land background. The model learns to rely on the background during training, and therefore makes mistakes when the background is swapped. CelebA contains images of celebrities' faces, and is known for its strong correlation between gender and hair color. Two popular language benchmarks are MultiNLI [Williams et al., 2018] and CivilComments-WILDS [Koh et al., 2021]. In MultiNLI, the task is to classify whether the second sentence is entailed by, neutral with, or contradicts the first sentence. The dataset contains a strong correlation between negation words and contradictions. CivilComments-WILDS is a dataset of online comments. The goal is to

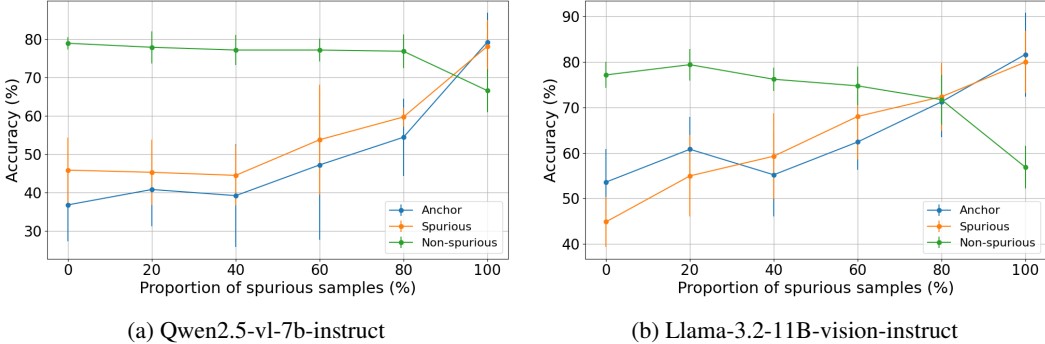

(a) Qwen2.5-vl-7b-instruct  (b) Llama-3.2-11B-vision-instruct

Figure 4: **Accuracy-robustness trade-off.** As the number of spurious correlations increases, both models' accuracies increase on both the anchor set and the spurious groups, and decrease on the non-spurious samples.

predict whether a comment is toxic, and it is correlated with the mention of certain demographic identities(e.g., male, female). These datasets focus on one task and contain only a few spuriously correlated attributes. Recent work introduces several synthetic datasets with multiple spurious correlations [Lynch et al., 2023, Joshi et al., 2023, Li et al., 2023b]. For example, UrbanCars is a dataset of urban and country cars, with both background and co-occurring objects as spurious attributes [Li et al., 2023b]. However, the number of spurious correlations and tasks in these benchmarks is still limited. These benchmarks emphasize a narrow task with foreknowledge of group and feature labels; we consider whether pretrained LVLMs are susceptible to spurious correlations in general settings.

Work most closely related to ours is MMSpuBench [Ye et al., 2024], a Visual Question Answering (VQA) benchmark that asks a model to choose which feature is best used to identify an object in an image. The choices include three spurious and one core feature. However, the faithfulness of the response is unclear: the model may still be using the core attribute to correctly identify the object despite selecting a spurious feature as its response. SpuriVerse shows that existing LVLMs make mistakes attributable to spurious correlations, achieving at best 35.0%.

**Methods for Mitigating Spurious Correlation** Previous approaches for mitigating spurious correlations typically require foreknowledge of correlated attributes [Sagawa et al., 2019, Sohoni et al., 2020, Zhang et al., 2022, Liu et al., 2021, Kirichenko et al., 2022]. Group DRO explicitly optimizes model performance on worst-case groups identified by spurious attributes [Sagawa et al., 2019]. Just-Train-Twice [Liu et al., 2021] and variants rely on two-stage pipelines where initial models identify errors or minority groups, which subsequently inform reweighting during retraining. Recent methods, like Concept Correction [Yang et al., 2024], utilize out-of-distribution examples to infer spurious attributes without requiring explicit labels. However, these methods generally assume a task-specific finetuning setting, limiting applicability to off-the-shelf LVLM deployments like ChatGPT. In contrast, we show that finetuning on diverse examples can generalize to unseen situations.

**Benchmarks for LVLMs** The recent proliferation of benchmarks for LVLMs [Fu et al., 2024, Yue et al., 2024, Saikh et al., 2022, Liu et al., 2024, Schwenk et al., 2022, Wen et al., 2023, Yao et al., 2025, Li et al., 2024b, Wu et al., 2025] focuses extensively on assessing capabilities in perception, reasoning, and knowledge across diverse domains. Concurrently, specialized benchmarks target critical LVLM shortcomings, such as hallucinations [Guan et al., 2024]. Our work uniquely investigates another fundamental vulnerability: susceptibility to generalized spurious correlations.

# 5   Conclusion

We introduce SpuriVerse, a benchmark for evaluating the susceptibility of large vision-language models to spurious correlations in real-world VQA tasks. By identifying and validating 124 distinct spurious patterns, SpuriVerse enables targeted evaluation of model behavior in the presence of dominant but unreliable correlations. Experiments across 15 LVLMs show that even state-of-the-art models frequently rely on such correlations. Fine-tuning on synthetic examples improves generalization to previously unseen spurious patterns, suggesting that models can learn to attend to broader scene

context. However, this improvement comes with a trade-off in overall performance, indicating that models may fundamentally rely on "shortcut" correlations. SpuriVerse supports future research on methods to promote robust scene understanding and reduce dependence on spurious cues. The limitations and societal impacts are discussed in the Appendix.

## Acknowledgments

This research was developed with funding from the Defense Advanced Research Projects Agency's (DARPA) SciFy program (Agreement No. HR00112520300). The views expressed are those of the author and do not reflect the official policy or position of the Department of Defense or the U.S. Government. This material is based upon work supported by the Defense Advanced Research Projects Agency and the Air Force Research Laboratory, contract number(s): FA8650-23-C-7316. Any opinions, findings and conclusions, or recommendations expressed in this material are those of the author(s) and do not necessarily reflect the views of AFRL or DARPA. We gratefully acknowledge gift and compute funding from Amazon and Google. Shangbin Feng would like to thank the support of the IBM PhD Fellowship and the Jane Street Graduate Research Fellowship.

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

This document supplements the main paper with additional details. Below is the outline:

- Section A details the prompts and annotation interface used for curating SpuriVerse.
- Section B details the prompts used, fine-tuning and evaluation details, and compute resources for experiments.
- Section C discusses the limitations of the work.
- Section D discusses the societal impacts of the work.
- Section E reports the licenses and terms of use for the models and datasets.
- Section F showcases some example images per category.
- Section G presents some images generated from various Stable Diffusion models.

# A SpuriVerse Details

## A.1 Prompts used for Dataset Curation

In this section, we include the prompts used for each step in the pipeline.

**Step 1: Select a challenging set.** The following template was used to prompt GPT-4o to find the error set on each source benchmark.

```
(System Prompt)

You will be given an image, and a multiple choice question regarding
the image.  You will provide your answer as one of the options (A), (B),
(C), or (D). You will answer correctly.  You will not use any fullstops
or punctuation.  You will not explain your answer or write words before
or after the answer.  Only the answer itself will you respond with.

(User Prompt)

<Image/>
Question:  {sample['question']}
Options:
(A) {sample['A']}
(B) {sample['B']}
(C) {sample['C']}
(D) {sample['D']}
Please select the correct answer from the options above.
```

**Step 2: Two-stage verification.** The following prompt was used to determine whether a sample falls into *VLM Accepted* subset.

```
(System Prompt)

You are a helpful assistant to determine if a model's error is caused
primarily by spurious correlations, patterns that can often be used to
predict the target, but are not actually causal.

(User Prompt)

<Image/>
Given this image, a Large Multi-modal Model was asked,
sample['question'], and given the choices:
(A) {sample['A']}
(B) {sample['B']}
```

```
(C) {sample['C']}
(D) {sample['D']}
The model chose {prediction} and the correct answer is {answer}.  The
error is most likely due to spurious correlation.  List the top two
spurious attributes that the model may have used to predict the wrong
answer {prediction}.
```

**Step 3: Generate counterfactual scene descriptions.** The following prompt was used to generate the pairs of counterfactual scene descriptions. Specifically, the goal is to first construct a description that enables the question-answer pair to hold in the scene description. Then, generate a spurious counterpart by extending the previous description to contain the spurious attributes provided.

```
<Image/>
You are given the following:
question:  {question}
answer:  {answer}
spurious attribute:  {attributes}

Based on the question and the answer, generate a description of a scene
such that when the question is asked, the answer is {answer}.  Keep the
description to one short sentence.

Write another one sentence description that includes the spurious
attribute while maintaining the same context.

Return the response in JSON format with the two keys:  "positive"
and "negative" where "positive" describes the scene with the spurious
attribute and "negative" describes the scene without the spurious
attribute.
```

**Step 5: Verify by Core vs. Spurious.** The same prompt in Step 1 was used to evaluate GPT-4o, Gemini 2.0 Flash, and Qwen-VL-Max on *Human Accepted* subset.

## A.2   Annotation Interface for Dataset Curation

Figure 5 displays the interface for refining spurious attributes and image descriptions. In step 1, the interface displays the image and question from the error set. In addition, it also displays GPT-4o's prediction and the ground truth answer. In step 2, annotators can view the prefetched spurious attributes by clicking on *From store* or generate new spurious attributes by clickong on *Run*. The annotators can then refine the spurious attributes in the text box. In step 3, annotators can click on *Run* and generate image descriptions for both spurious and core groups based on the spurious attributes extracted in Step 2. In step 4, the images for spurious and core groups can be generated based on the descriptions in step 3. Annotators can go back to step 3 and refine the descriptions if the images generated are not faithful. In step 5, annotators can take a peek at models' evaluations on one image for spurious group and one image for core group.

## A.3   Prompt used for Category Identification

```
<spurious examples.csv/>

You are given a list of examples that a model answers incorrectly due
to spurious correlation.  For each row, a model makes an error due to
the correlation between the spurious attribute (spuriousAttr) and the
prediction.  If the prediction is 'Yes' or 'No', refer to the question
for context.
```

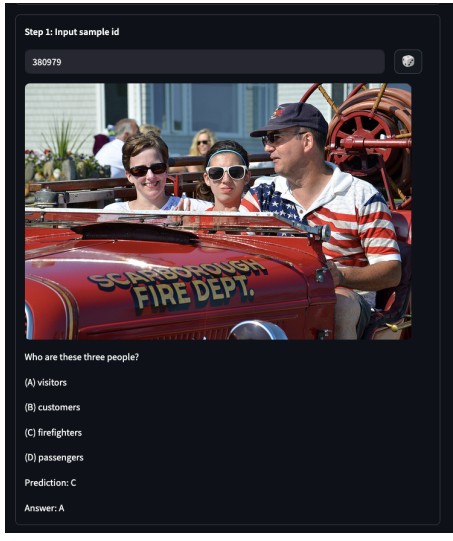

(a) Step 1

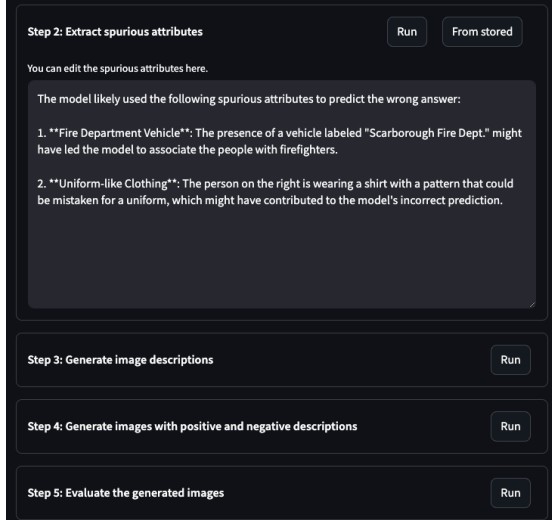

(b) Steps 2-5

Figure 5: **Annotation interface** for refining spurious attributes and image descriptions. The curation pipeline consists of 5 steps: Step 1 displays the image and question from the error set. Step 2 allows annotators to view extracted spurious attributes by clicking on *From store* or generating new spurious attributes by clicking on *Run*. Annotators can edit the spurious attributes. Step 3 generates image descriptions based on the extracted or edited spurious attributes. Step 4 then generates images for spurious and core groups. Annotators can go back to step 3 and refine the description if the images generated are not faithful. Step 5 allows annotators to take a peek at models' evaluations on a few images (One for each group).

```
Determine the spurious correlation for each row.  Then create a
taxonomy based on the spurious correlations.
```

## A.4  Dataset Cost

We used Stable Diffusion to generate the synthetic images for verifying the spurious correlations, and the spurious groups also make up part of the dataset. In our curation pipeline, 194 samples were being selected after the human-VLM verfication step. We thus generated $194 \times 2 \times 10 = 3880$ images using Stable Diffusion Ultra. Since generating each image costs 8 credits, and each credit costs \$0.01, the total cost is $3880 \times 8 \times 0.01 = 310.4 dollars$.

## B   Experiments

### B.1   Prompts used for Main results

We used the following prompt:

```
(System Prompt)

You will be given an image, and a multiple choice question regarding
the image.  You will provide your answer as one of the options (A), (B),
(C), or (D). You will answer correctly.  You will not use any fullstops
or punctuation.  You will not explain your answer or write words before
or after the answer.  Only the answer itself will you respond with.
```

```
(User Prompt)

<Image/>
Question:  {sample['question']}
Options:
(A) {sample['A']}
(B) {sample['B']}
(C) {sample['C']}
(D) {sample['D']}
Please select the correct answer from the options above.
```

## B.2   Prompts used for Prompting strategies

We kept the user prompt the same as Main Results. For Chain-of-thought, we used the following system prompt:

```
(System Prompt)

You will be given an image, and a multiple choice question regarding
the image.  Think step by step and give a final answer.  You will
include one of the choices (A), (B), (C), or (D) in your final answer.
```

For Spurious Aware, we used the following sytem prompt:

```
(System Prompt)

You will be given an image, and a multiple choice question regarding
the image.  Be aware that there may be some spurious features in the
image that associate with some of the options.  Describe the potential
spurious features.  Then give a answer without using the spurious
features.  You will include one of the choices (A), (B), (C), or (D)
in your final answer.
```

## B.3   Models

We evaluate SpuriVerse on 15 recent LVLMs, including GPT-4o [OpenAI, 2023], GPT-4o-mini [OpenAI, 2023], o4-mini [OpenAI, 2023], o3 [OpenAI, 2023], Gemini 2.0 Flash [DeepMind, 2024], Gemini 1.5 Pro [DeepMind, 2024], Claude 3.7 Sonnet [Anthropic, 2024], Qwen-VL-Max [Cloud, 2025], Qwen-VL-Pro [Cloud, 2025], Qwen2.5-vl-7b-instruct [Team, 2025], Qwen2.5-vl-32b-instruct [Team, 2025], Llama-3.2-11B-vision-instruct [Grattafiori et al., 2024], Llama-3.2-90B-vision-instruct [Grattafiori et al., 2024], LLaVA-v1.6 7b [Liu et al., 2023a] and LLaVA-v1.5 [Liu et al., 2023b].

We used OpenAI API [OpenAI, 2023] for making requests to GPT-4o, GPT-4o-mini, o4-mini, o3. We used Gemini API [DeepMind, 2024] for making requests to Gemini 2.0 Flash, Gemini 1.5 Pro. We used Anthropic API [Anthropic, 2024] for making requests to Claude 3.7 Sonnet. We used Qwen API [Cloud, 2025] for making requests to Qwen-VL-Max, Qwen-VL-Pro. We accessed Qwen2.5-vl-7b-instruct, Qwen2.5-vl-32b-instruct, Llama-3.2-11B-vision-instruct, Llama-3.2-90B-vision-instruct via Unsloth AI [Daniel Han and team, 2023]. We accessed LLaVA-v1.6 7b and LLaVA-v1.5 via Hugging Face [Wolf et al., 2020].

**Hyperparameters** During evaluation, for the reasoning models (o3 and o4-mini), we set max_tokens to 3000. For all other models, we set max_tokens to 300. All the open-sourced models are 4-bit quantized during evaluation.

**Versions** For GPT-4o, we used version "gpt-4o-2024-08-06". For GPT-4o-mini, we used version "gpt-4o-mini-2024-07-18". For Claude 3.7 Sonnet, we used version "claude-3-7-sonnet-20250219".

## B.4 Finetuning details

We divided both the anchor set and the spurious groups into train/val/test sets according to the ratio of 70/10/20. We finetuned Llama-3.2-11B-vision-instruct and Qwen2.5-vl-7b-instruct on the train and val sets of anchors and spurious groups, respectively, and evaluated on the test sets. As a baseline, we also considered the "non-spurious set", which was sampled randomly from the source benchmarks. The samples are drawn with the same benchmark distribution as the anchor set. Similarly, the non-spurious set is further split according to the ratio of 70/10/20. We also finetuned on the "Mixed set", which was the concatenation of spurious groups and the "non-spurious set".

All finetuning results were measured across 5 seeds, where each seed corresponds to a different split of the data.

We finetuned both Llama-3.2-11B-vision-instruct and Qwen2.5-vl-7b-instruct using unsloth [Daniel Han and team, 2023].

We used the following hyperparameters for both models:

```
finetune_vision_layers=True,
finetune_language_layers=True,
finetune_attention_modules=True,
finetune_mlp_modules=True,
r=16,
lora_alpha=16,
lora_dropout=0,
bias="none",
random_state=3407,
use_rslora=False,
loftq_config=None
```

We used the following SFT configuration for both models:

```
per_device_train_batch_size=2,
gradient_accumulation_steps=4,
warmup_steps=5,
num_train_epochs=10,
learning_rate=2e-4,
logging_steps=1,
optim="adamw_8bit",
weight_decay=0.01,
lr_scheduler_type="linear",
seed=3407,
remove_unused_columns=False,
dataset_text_field="",
dataset_kwargs={"skip_prepare_dataset":  True},
dataset_num_proc=4,
max_seq_length=2048,
eval_strategy="epoch",
load_best_model_at_end=True,
save_strategy="best",
metric_for_best_model="eval_loss",
greater_is_better=False,
save_total_limit=2,
```

We used the following instruction format during finetuning:

```
<Image/>
Question:  sample['question']
Options:
(A) sample['A']
(B) sample['B']
(C) sample['C']
(D) sample['D']
Please select the correct answer from the options above.
```

## B.5  Robustness-accuracy Tradeoff

The procedure to replace spurious samples with non-spurious samples is described in Algorithm 1. Particularly, we use the anchor set $\mathcal{S}_{\text{anchor}}$ as the reference to decide the distribution of the source benchmarks in the training set. If a sample $s_i$ in $\mathcal{S}_{\text{anchor}}$ is determined to use non-spurious samples, then we randomly sample 10 images from the source benchmark $s_i$ originates from. Otherwise, we use the original spurious group images $G_i$ of size 10.

As a toy example, suppose $\mathcal{S}_{\text{anchor}} = \{s_1, s_2, s_3, s_4, s_5\}$. Let $\{s_1, s_2, s_3, s_4\}$ be the training split at first, and each of them comes from a distinct benchmark we use. Let the second split between spurious and non-spurious yields $\mathcal{S}_{\text{train, spurious}} = \{s_1, s_2\}, \mathcal{S}_{\text{train, non-spurious}} = \{s_3, s_4\}$ with $r = 50\%$. Then, the next step will yield a training set of 40 samples, consisting of 20 samples from $G_1, G_2$, 10 samples drawn from the source benchmark of $s_3$, and 10 samples drawn from the source benchmark of $s_4$. With this training set $\mathcal{S}'_{\text{train}}$, we then do a train-val split with a fraction of 87.5% and 12.5%.

---

**Algorithm 1** Sampling procedure to replace spurious samples with non-spurious samples

---

**Input:** anchor set $\mathcal{S}_{\text{anchor}}$
**Input:** spurious fraction $r$
**Input:** spurious group samples $G$       $\triangleright$ $G_i$ is a set of 10 samples using synthetic images for $s_i \in \mathcal{S}_{\text{anchor}}$
**Input:** benchmark samples $B$       $\triangleright$ $B_i$ is the set of all samples from the $i$-th benchmark

Let $b(s)$ be the source benchmark of a sample $s$
$\mathcal{S}_{\text{train}}, \mathcal{S}_{\text{test}} \leftarrow \texttt{random\_split}(\mathcal{S}_{\text{anchor}}, \texttt{fraction=[0.8, 0.2]})$
$\mathcal{S}_{\text{train, spurious}}, \mathcal{S}_{\text{train, non-spurious}} \leftarrow \texttt{random\_split}(\mathcal{S}_{\text{train}}, \texttt{fraction=[}r, 1-r\texttt{]})$
$\mathcal{S}'_{\text{train}} \leftarrow \{\}$
**for** $s_i \in \mathcal{S}_{\text{train, spurious}}$ **do**
    add $G_i$ to $\mathcal{S}'_{\text{train}}$
**end for**
**for** $s_i \in \mathcal{S}_{\text{train, non-spurious}}$ **do**
    $j \leftarrow b(s_i)$
    $S_{s_i, B_j} \sim B_j$ where $|S_{s_i, B_j}| = 10$       $\triangleright$ Draw 10 random samples from the source benchmark
    add $S_{s_i, B_j}$ to $\mathcal{S}'_{\text{train}}$
**end for**
$\mathcal{S}''_{\text{train}}, \mathcal{S}''_{\text{val}} \leftarrow \texttt{random\_split}(\mathcal{S}'_{\text{train}}, \texttt{fraction=[0.875, 0.125]})$       $\triangleright$ fraction to make train/val/test split 70/10/20 w.r.t. $\mathcal{S}_{\text{anchor}}$

**Output:** $\mathcal{S}''_{\text{train}}, \mathcal{S}''_{\text{val}}, \mathcal{S}_{\text{test}}$

---

## B.6  Compute Resources

The experiments were conducted on a 4xNVIDIA H100, where the GPU memory is 4x95830MB. The CPU architecture is x86_64, and there are 64 CPUs.

For the finetuning experiments, since both the Llama-3.2-11B-vision-instruct and Qwen2.5-vl-7b-instruct are 4-bit quantized and optimized with UnslothAI, approximately 12GB of GPU memory is sufficient for finetuning. Finetuning each model on spurious groups/non-spurious groups for 10 epochs takes about 3 hours on a single H100. Overall, the total compute time is $num\_setups(3) \times$

$num\_models(2) \times num\_seeds(5) \times duration(3) = 90hours$. For the accuracy-robustness trade-off experiment, the total compute time is $num\_setups(6) \times num\_models(2) \times num\_seeds(5) \times duration(3) = 180hours$. In this case, the setup refers to the proportion of the spurious samples.

Finetuning on the anchor set takes approximately 0.5 hour, since the anchor set is a much smaller set. Hence the total compute time is $num\_setups(3) \times num\_models(2) \times num\_seeds(5) \times duration(0.5) = 15hours$.

For the main results, running Llama-3.2-90B-Vision-Instruct for inference takes about 45 GB GPU memory, and running Qwen2.5-vl-32b-instruct takes about 24 GB GPU memory. All the other open-source models can be run under 12 GB of GPU memory. Evaluation of each non-reasoning model takes about 0.5 hour on spurious and non-spurious samples. Hence, the total compute time is $num\_setups(2) \times num\_models(13) \times duration(0.5) = 13hours$. Evaluation of the reasoning models (o3 and o4-mini) takes about 10 hours. Hence, the total compute time is $num\_setups(2) \times num\_models(2) \times duration(10) = 40hours$.

Evaluating on the anchor set takes about 0.05 hours for the non-reasoning models, and 1 hour for the reasoning models. Hence, the total compute time for non-reasoning models is $num\_setups(2) \times num\_models(13) \times duration(0.05) = 1.3hours$, and the total compute time for reasoning models is $num\_setups(2) \times num\_models(2) \times duration(1) = 4hours$.

The full research project does not require more compute than the experiments reported in the paper.

## C   Limitations

**Curation Bias.** Two forms of bias can exist in our curation pipeline. First, we use GPT-4o as the single strong model in the early steps of the curation pipeline. This procedure can limit us to spurious correlations closer to its training distribution. However, our last counterfactual verification attempts to mitigate this potential bias. Secondly, the human annotations were done by two contributors on the team with agreement. There can be annotation variance when it comes to other annotators.

**Sample Format.** In our work, we evaluate models on multiple-choice VQA questions, so that the target choice associated with the spurious feature is known ahead of time. This setup limits the output space compared to open-generation formats, allowing easier investigation into whether a model's prediction is driven by spurious correlations. The open-generation format, while offering greater expressive potential, introduces both opportunities and challenges. On one hand, it could reveal cases where spurious features exist but the corresponding target concept does not appear in any of the predefined multiple-choice options, thus uncovering new forms of spurious associations. On the other hand, evaluating open-generation outputs requires either human intervention or automatic judgment (e.g., via LLM-as-a-judge), both of which may introduce additional biases and complexity. As ongoing work continues to explore this evaluation layer and its potential pitfalls, we leave this extension to open-generation settings for future work.

## D   Societal Impacts

While our work serves as a new spurious correlation benchmark for LVLMs, it is not to be used as a bullet-proof shield to claim that "a model with good accuracy on SpuriVerse can be free from all potential spurious correlation attacks," and thus reduce the efforts in improving the robustness of these models. As we also release the scene descriptions for the spurious group generation, malicious users can potentially design attacks more easily to generate harmful images while maintaining their usefulness to improve robustness against spurious correlation evaluated on SpuriVerse, causing a false promise of "better" models. In our work, we demonstrate the possibility of generalizing to unseen spurious correlations when finetuning on a diverse set of spurious correlations. We hope SpuriVerse can help foster more future work in this direction. We believe that SpuriVerse provides valuable insights into LVLMs' robustness to common spurious correlations when used properly.

## E   Licenses

The anchor set of SpuriVerse is collected from AOKVQA [Schwenk et al., 2022], SEEDBench [Li et al., 2023a], SEEDBench2 [Li et al., 2024b], NaturalBench [Li et al., 2024a].

The license or terms of use for each dataset and model is provided in the following:

Datasets:
AOKVQA: Apache-2.0.
SEEDBench: Attribution-NonCommercial 4.0 International.
SEEDBench2: Attribution-NonCommercial 4.0 International.
NaturalBench: Apache-2.0.

Models: GPT-4o: OpenAI's Term of Use and Business Terms
GPT-4o-mini: OpenAI's Term of Use and Business Terms
o4-mini: OpenAI's Term of Use and Business Terms
o3: OpenAI's Term of Use and Business Terms
Gemini 2.0 Flash: Google's API Terms of Service
Gemini 1.5 Pro: Google's API Terms of Service
Claude 3.7 Sonnet: Anthropic's Terms of Service
Qwen-VL-Max: Alibaba Cloud's Terms of Service
Qwen-VL-Pro: Alibaba Cloud's Terms of Service
Qwen2.5-vl-7b-instruct: Apache-2.0
Qwen2.5-vl-32b-instruct: Apache-2.0
Llama-3.2-11B-vision-instruct: Llama 3.2 Community License
Llama-3.2-90B-vision-instruct: Llama 3.2 Community License
LLaVA-v1.6 7b: LLAMA 2 Community License
LLaVA-v1.5: LLAMA 2 Community License

## F Category Exemplar Images

The following are representative examples of each spurious correlation category.

## G Images Synthesized From Stable Diffusion Ultra, Core, and 3.5-medium

The following are example images generated using Stable Diffusion Ultra, Core, and 3.5-medium models.

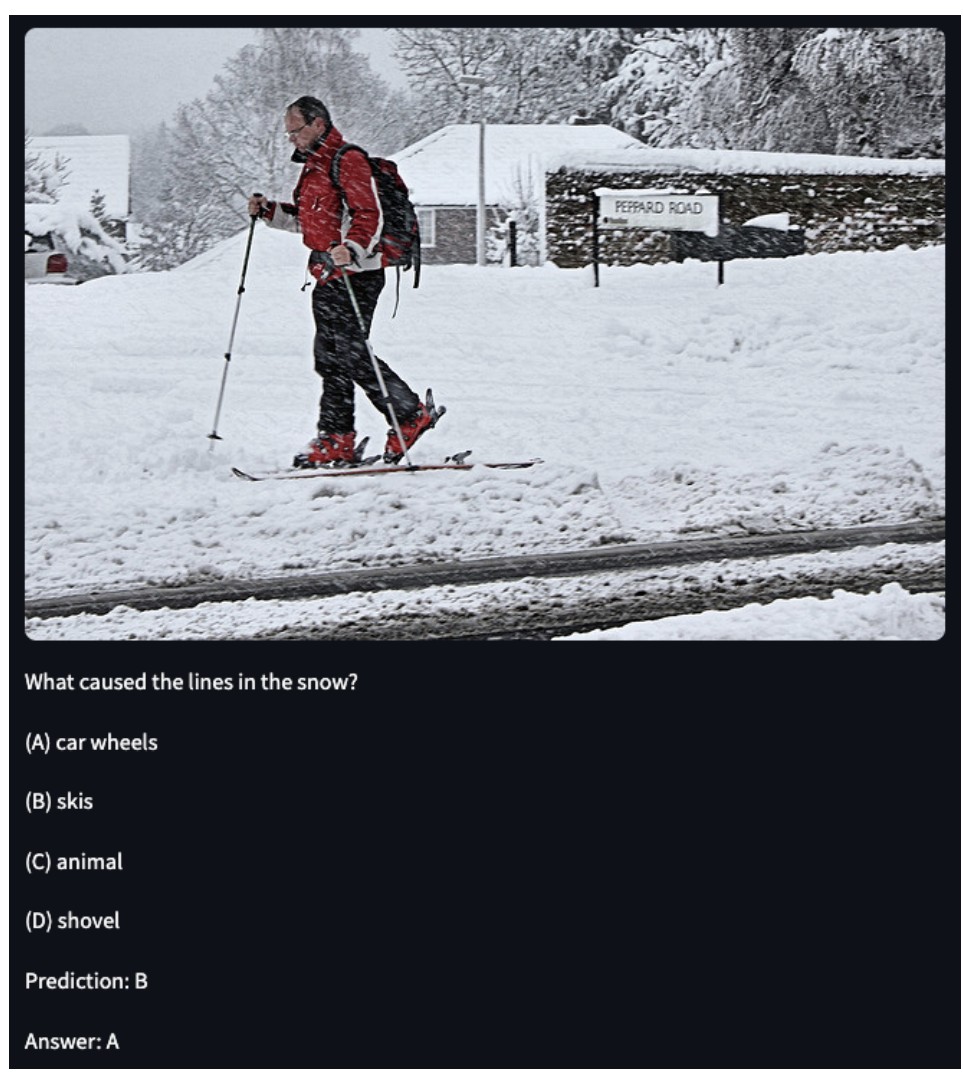

Figure 6: Object Co-Occurrences

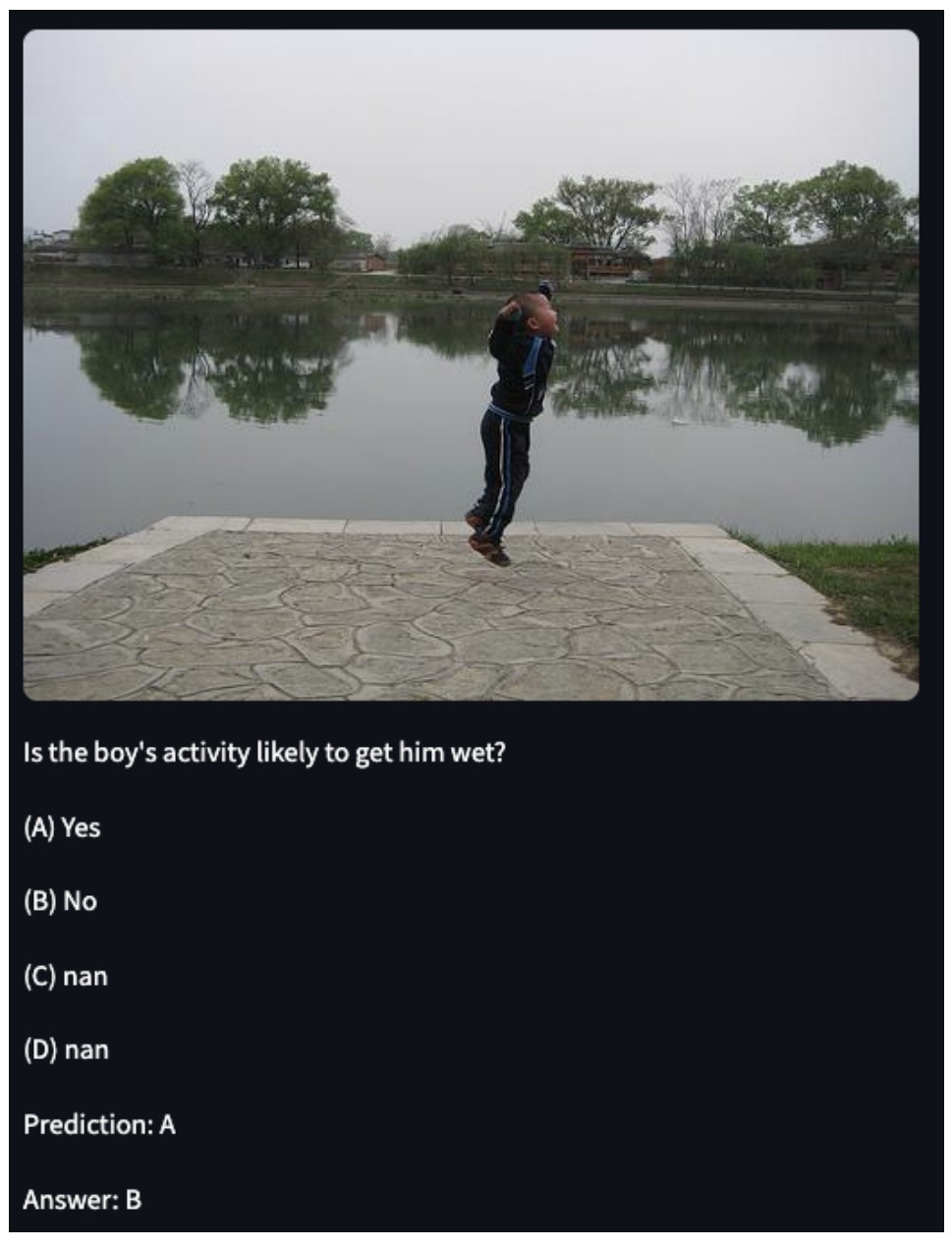

Figure 7: Context Cues

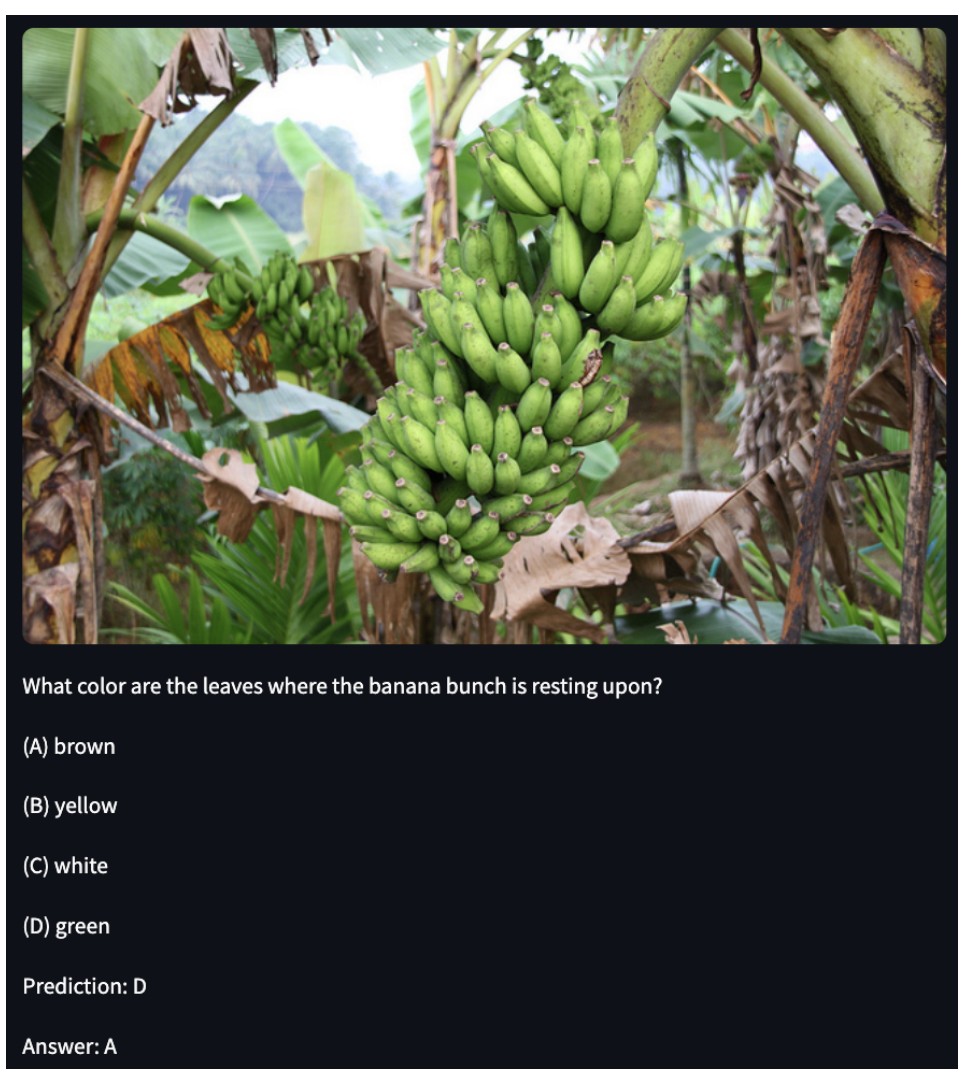

Figure 8: Visual Predominance

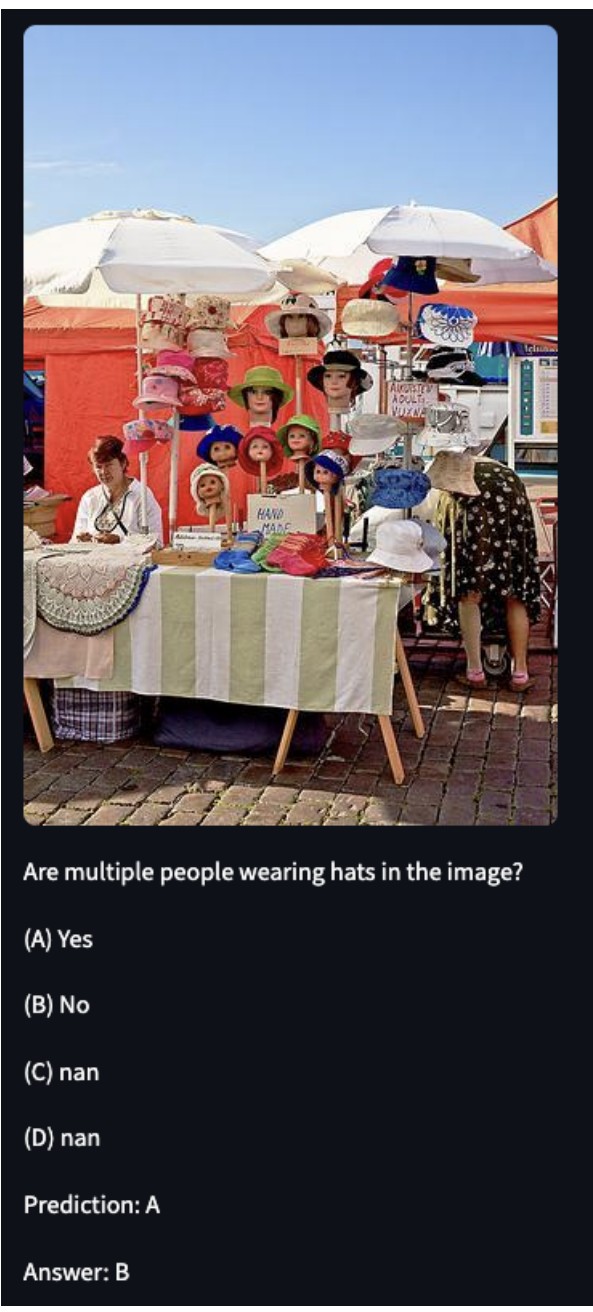

Figure 9: Visual Resemblance

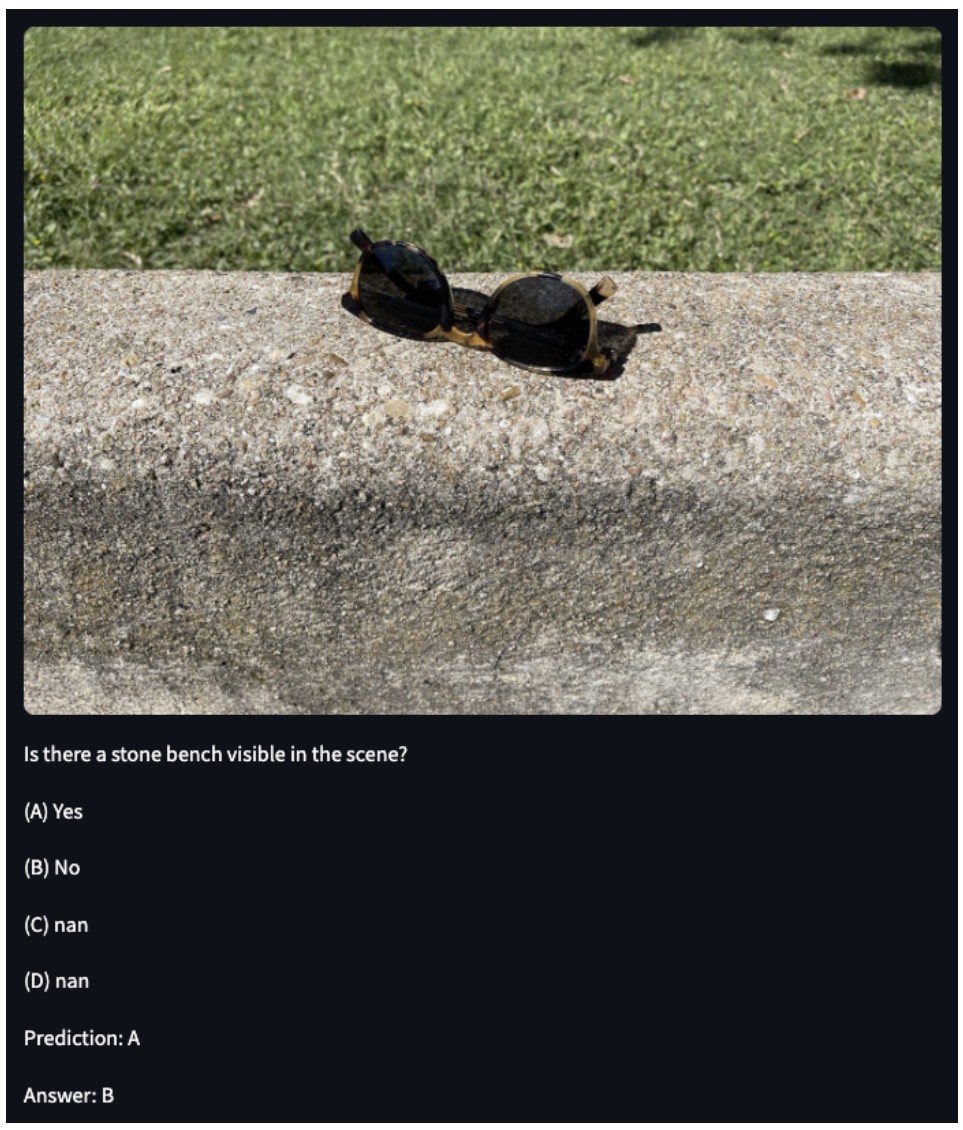

Figure 10: Physical Properties

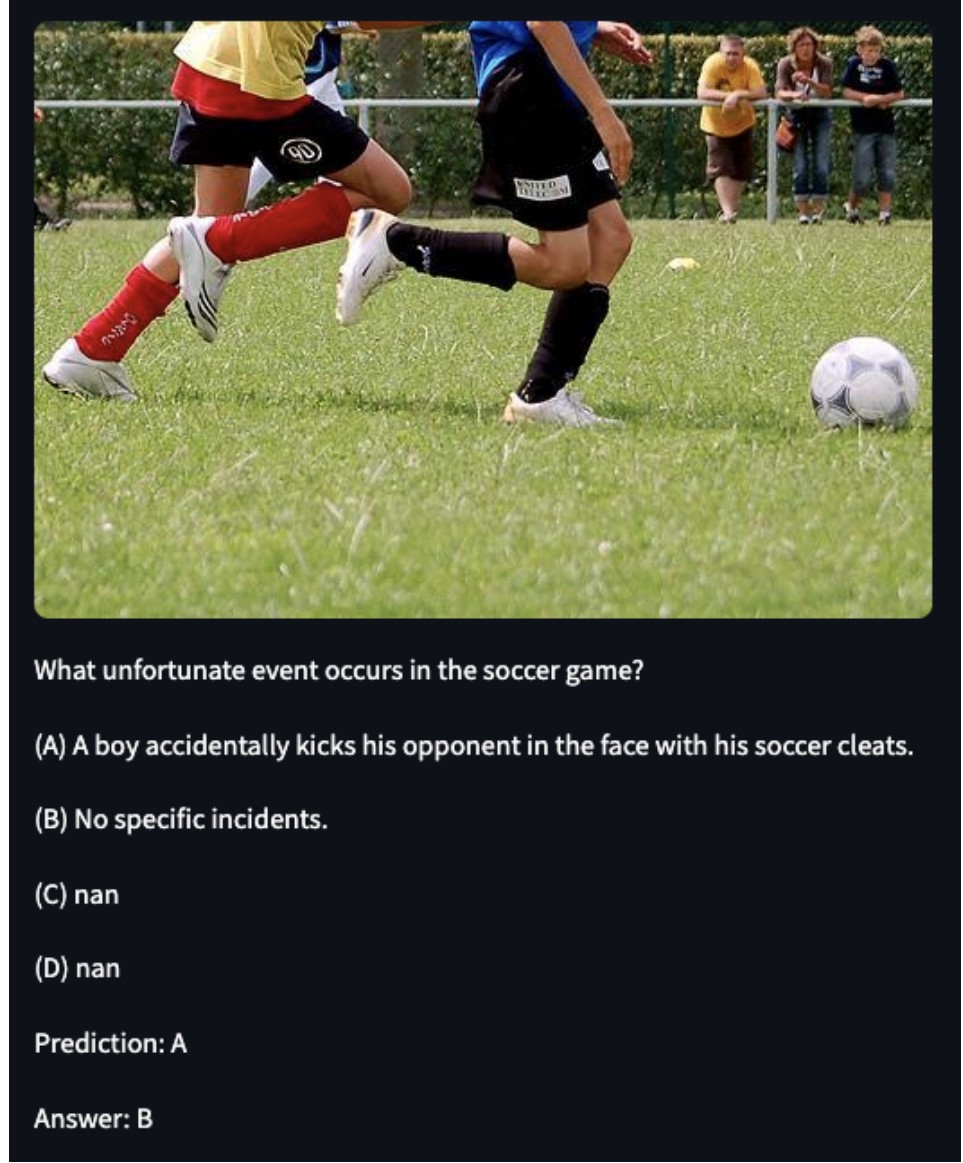

What unfortunate event occurs in the soccer game?

(A) A boy accidentally kicks his opponent in the face with his soccer cleats.

(B) No specific incidents.

(C) nan

(D) nan

Prediction: A

Answer: B

Figure 11: Spatial Relationships

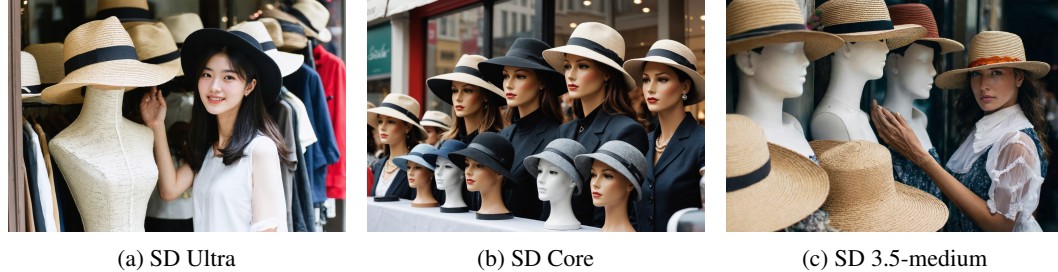

(a) SD Ultra      (b) SD Core      (c) SD 3.5-medium

Figure 12: Prompt: A lady is selling hats, which are placed on mannequins.

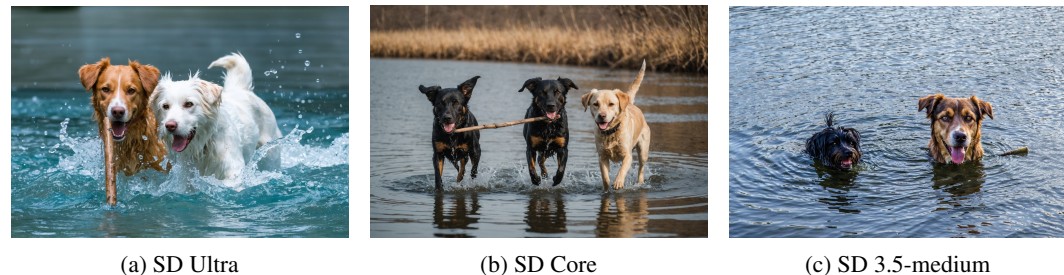

(a) SD Ultra  (b) SD Core  (c) SD 3.5-medium

Figure 13: Prompt: Two dogs are in water. A stick is present.

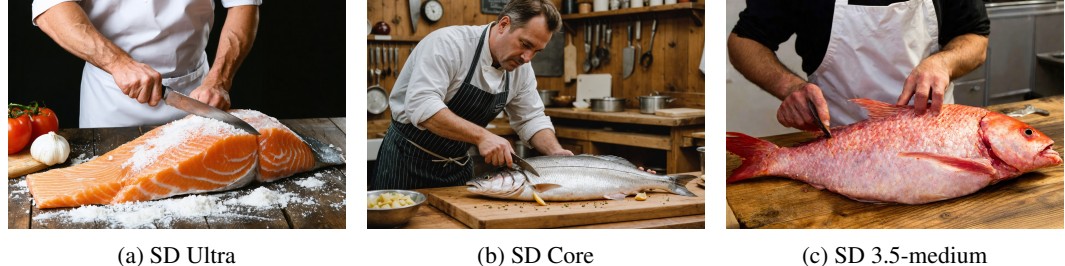

(a) SD Ultra  (b) SD Core  (c) SD 3.5-medium

Figure 14: Prompt: A butcher wearing a white apron is cutting a large fish on a wooden table.

