# OpenReview forum: "Escaping the SpuriVerse: Can Large Vision-Language Models Generalize Beyond Seen Spurious Correlations?"
_NeurIPS.cc/2025/Datasets_and_Benchmarks_Track — NeurIPS 2025 Datasets and Benchmarks Track poster_

### Official Review · Reviewer_VWxi · 2025-06-17

**Rating:** 4
**Confidence:** 4

**Summary:**

This submission studies spurious correlations in the context of large vision-language models (LVMs). It hypothesizes that correlations between spurious features and target concepts can persist and compromise LVMs’ performance. To this end, a benchmark is proposed to evaluate susceptibility to spurious correlations in generalized settings. The focus is on dominant correlations that are not necessarily irrelevant but are not generally reliable. The experiments cover most of the LVMs and show that allLVLMsperformworsethanrandomguessontheanchorset. Futher, the spurious-aware prompt is shown to be effective in improving the model accuracy. Moreover, the experiments indicate fine-tuning on some categories of spurious correlation can be generalized to other categories.

**Additional Feedback:**

- "Finetuning can cause spurious correlations to arise between non-essential features and the target labels, but benchmarks to study these effects involve contrived settings and narrow tasks". The first sentence in the Abstract is very confusing. It makes the reader think this submission will study the spurious correlation caused by fine-tuning LVMs. Simply introducing the definition of spurious correlation is fine.

**Dataset Code Accessibility:**

Partly

**Dataset Code Comments:**

The datasets are available; however, it would be better to include the code for using the dataset for evaluation.

**Ethical Considerations:**

No, there are no or only very minor ethics concerns

**Final Justification:**

Thank you for the detailed response, which has addressed all of my concerns. I have also carefully read the comments from the other reviewers. I maintain my original rating of Borderline  Accept.

**Limitations Weaknesses:**

- [***Potential Effect of Synthetic Artifacts***] Since there are synthetic images (Step 4), it would be helpful to discuss the failure cases of the generated images. For example, generated images may contain imperceptible noise, which could affect the performance of LVMs.
- [***Threshold choice***] Step 5 needs some clarification: why is a 30% difference in accuracy used?

     [***Category clarity:***] Also, it would be better to include sample images for each category in Section 2.3.
- [***Discuss Open-Generation***] In the limitations section, the submission mentions the open-generation format. Please add more details on this point.

**Strengths Contributions:**

- The motivation is sound. It is useful to study the potential spurious correlations in the context of LVMs. The Introduction (lines 44–57) clearly presents the motivation and the difficulty of constructing a benchmark.
- The pipeline (Section 2.2) for constructing the benchmark is reasonable: building upon the errors of LVMs on existing benchmarks is a good approach. Figures 1 and 2 nicely introduce the idea of SpuriVerse. (It would be helpful to increase the resolution of both figures.)
- The experiments are extensive and well organized. Prompting, fine-tuning, and the accuracy–robustness trade-off are well presented. Each of them provides insights into the capabilities and limitations of LVMs.

---

> ### Author Rebuttal · Authors · 2025-07-31
>
> We thank the reviewer for their thoughtful comments. We are glad that the reviewer recognized that the problem is important, the curation pipeline is sound, and the experiments are extensive and well-organized.
>
> > [Potential Effect of Synthetic Artifacts] Since there are synthetic images (Step 4), it would be helpful to discuss the failure cases of the generated images. For example, generated images may contain imperceptible noise, which could affect the performance of LVMs.
>
> We appreciate the point on failure cases of the generated images. We have briefly discussed this in line 120-123. The generated images tend to involve hallucinations such that they become unfaithful to the text descriptions. Therefore, we use human annotators to examine the generated images and modify the text descriptions to improve faithfulness. The images are further reviewed by the annotators so that no harmful contents are present.
>
> Regarding the imperceptible noise, when verifying the spurious correlations in step 5, we evaluate the VLMs on both spurious and core groups, which are both synthetic images. This ensures a fair comparison, so that the presence of the spurious feature is the primary difference among the two groups.
>
> > [Threshold choice] Step 5 needs some clarification: why is a 30% difference in accuracy used?
>
> The primary goal of step 5 is to validate that the spurious attribute indeed contributes to the model’s prediction. The % difference approximates how strong is the spurious correlation captured by the model. We selected 30% because it gives us a good balance between ensuring samples with strong correlations and a large enough dataset, which allows us to evaluate the generalizability of models when finetuning on diverse spurious correlations . We will release the core groups in the dataset, so that researchers can control the degree of correlation and select a tighter set if needed.
>
> > [Category clarity:] Also, it would be better to include sample images for each category in Section 2.3.
>
> We thank the suggestion and will include sample images for each category in the Appendix.
>
> > [Discuss Open-Generation] In the limitations section, the submission mentions the open-generation format. Please add more details on this point.
>
> In our work, we evaluate models on multiple-choice VQA questions, so that the target choice associated with the spurious feature is known ahead of time. This approach allows for easier investigation into whether a prediction is made due to spurious correlation.  The open-generation format introduces both potential and challenges. On one hand, it could provide a possibility where the spurious features exist but the corresponding target doesn’t appear in any of the existing multiple-choice options. Open-generation might reveal these spurious correlations. On the other hand, evaluating open-generation requires either human intervention or an automatic approach such as LLM-as-a-judge. This layer of evaluation might introduce additional biases and complexity. Therefore, we leave this exploration for future work.
>
> > The datasets are available; however, it would be better to include the code for using the dataset for evaluation.
>
> We included the code for evaluating VLMs on the dataset in the GitHub repo. The evaluation guideline is documented in the Adhoc Evaluation section.
>
> > "Finetuning can cause spurious correlations to arise between non-essential features and the target labels, but benchmarks to study these effects involve contrived settings and narrow tasks". The first sentence in the Abstract is very confusing. It makes the reader think this submission will study the spurious correlation caused by fine-tuning LVMs. Simply introducing the definition of spurious correlation is fine.
>
> We appreciate the suggestion and will revise the abstract accordingly.

---

> > ### Comment · Reviewer_VWxi · 2025-08-05
> >
> > Thank you for the detailed response, which has addressed all of my concerns. I have also carefully read the comments from the other reviewers. I maintain my original rating of Borderline Accept.

---

### Official Review · Reviewer_Ho3E · 2025-07-01

**Rating:** 5
**Confidence:** 4

**Summary:**

Finetuning can lead to models learn spurious correlations. These spurious correlations can compromise models' performance and generalization. However, existing Large Vision-Language Models' (LVLMs) evaluation suites do not examine whether zero-shot LVLMs rely on spurious features to make incorrect predictions. In this paper, the authors propose a benchmark to evaluate an LVLM's susceptibility to spurious correlations in generalized settings. To construct the benchmark, the authors introduce a pipeline to identify and validate the spurious patterns. The authors evaluate 15 open and closed-source LVLMs on SpuriVerse, finding that existing methods are vulnerable to spurious correlations.

**Additional Feedback:**

Some typos:

Line 73-74: "such as as Chain-of-Thought". Repeated "as".

Line 197: "and and slight performance decline". Repeaded "and".

Line 208: "Its accuracy drops 66.48% when finetuned on anchors.". It is unclear (drops by 66.48% vs drops to 66.48%).

**Dataset Code Accessibility:**

Yes

**Dataset Code Comments:**

The dataset is available on Huggingface. The code is also available on Github.

**Ethical Considerations:**

No, there are no or only very minor ethics concerns

**Final Justification:**

The authors propose a benchmark to evaluate an LVLM's susceptibility to spurious correlations in generalized settings. 124 distinct types of spurious correlations are included in SpuriVerse. Authors' rebuttal has addreessed my concerns.

**Limitations Weaknesses:**

1. The font sizes in Figures 1 and 2 are too small. The resolution is low, making them difficult to read. It is recommended that the authors use vector graphics or embed high-resolution PDF images into the paper.
2. The number of unique samples of SpuriVerse remains relatively small (only 124 anchor examples and their synthetic counterparts). This limited scale raises concerns about the generalizability of the conclusions drawn in the paper. For example, it is unclear whether the observed trade-off between robustness and performance would persist when fine-tuning is conducted on larger and more diverse sets of spurious examples.

**Strengths Contributions:**

1. The authors introduce the first spurious correlation benchmark for LVLMs evaluation. This benchmark covers 124 distinct types of spurious correlations extracted from real-world datasets.
2. The authors analyze the performance of 15 open and closed-source LVLMs on SpuriVerse. They found that even the best model achieves the highest accuracy of 37.9%, lagging behind a random guess by 3.2% on the anchor set.
3. The authors find a trade-off between performance and robustness. They show that finetuning on synthetic spurious samples can improve robustness to unseen spurious correlations, but often degrade performance on non-spurious examples.

---

> ### Author Rebuttal · Authors · 2025-07-31
>
> We thank the reviewer for their thoughtful comments. We are glad that the reviewer recognized that the benchmark is novel and the results are interesting.
>
> > The font sizes in Figures 1 and 2 are too small. The resolution is low, making them difficult to read. It is recommended that the authors use vector graphics or embed high-resolution PDF images into the paper.
>
> We thank the reviewer for the recommendation; we will increase the font sizes in figures 1 and 2, and embed high-resolution PDF images into the final paper.
>
> > The number of unique samples of SpuriVerse remains relatively small (only 124 anchor examples and their synthetic counterparts). This limited scale raises concerns about the generalizability of the conclusions drawn in the paper. For example, it is unclear whether the observed trade-off between robustness and performance would persist when fine-tuning is conducted on larger and more diverse sets of spurious examples.
>
> The dataset may indeed seem small if we consider each sample as a single data point. However, rather than finding merely 124 instances of spurious correlations, we show via step 5 in the pipeline that these are 124 patterns that consistently lead VLMs to fail. Specifically, for each pattern, when the spurious feature is present (i.e., the spurious group), the VLMs tend to make errors much more often than when it is not (i.e., the core group). In comparison, existing popular spurious correlation datasets only contain 1-2 spurious patterns, while having over tens or even hundreds of thousands of examples (E.g., Waterbirds has 11,788 samples, CelebA has 202,599 samples). The design of our curation pipeline allows us to easily expand the dataset by generating more images per correlation. Our goal is to significantly generalize the study of spurious correlations by including over 100x more patterns than previous benchmarks.
>
> We appreciate the reviewer's attention to details and will address all identified typos in future manuscripts.

---

> > ### Comment · Reviewer_Ho3E · 2025-08-03
> >
> > Thanks for the rebuttal. The authors' rebuttal has address my concerns. Thus, I decide to raise my score.

---

### Official Review · Reviewer_hS21 · 2025-07-01

**Rating:** 5
**Confidence:** 4

**Summary:**

The paper introduces spuriverse a benchmark/dataset for benchmarking spurious correlations in VLMs. The paper uses GPT-4o for to initially identify errors in question answering on a "challenging" dataset  and then asks the GPT 4.0 with another disconnected prompt to verify whether the error was caused due to spurious correlations and is asked to generate plausible spurious correlations, which is then human verified. A Stable Diffusion model is used to generate synthetic datasets with 20 images, 10 with spurious correlations (spurious group) and 10 without spurious correlations(core group) guided by the anchor (real image), to description to produce the said images are first computed, 1 with the spurious attribute present and 1 without mentioning the spurious correlation for the spurious and core group respectively. The accuracy difference between the core group and spurious group is computed for various popular VLMs and found that, the core group outperforms the spurious group, suggesting that the spurious correlation dataset generally provides valid spurious correlation, although only a subset of spurious correlations. The paper also benchmark the effectiveness of different prompting techniques on spurious correlations, chain of thoughts possibly due to its reasoning has lesser error due to spurious correlations compared to direct prompting. The paper then introduced a spurious aware prompting technique, that explicitly alerted the VLM of possible spurious correlations, which had better improvements than chain of thoughts.  The paper then explores finetuning VLMs on the generated spurious dataset, which increased accuracy on questions from the spurious group but the accuracy dropped on non spurious group. A mixed set of spurious and non-spurious samples balanced the tradeoff.

**Additional Feedback:**

I hope to have given feedback in weakness section

**Dataset Code Accessibility:**

Yes

**Dataset Code Comments:**

I was able to load the dataset on huggingface on google colab after updating my datasets to 3.6.0 using pip install -U datasets. The description for the dataset was clear and detailed. The code for benchmark is provided in github

**Ethical Considerations:**

No, there are no or only very minor ethics concerns

**Final Justification:**

Upon reading the rebuttal, I think the design choices are justified. I also accept that the in depth spurious correlation dataset could be a future work.

**Limitations Weaknesses:**

1. **Stable Diffusion Bias**: The depth of the spurious labels are to be explored, VLM A might associate a red apple as a spurious correlation, VLM B with green apples, if the spurious label generated by GPT 4o is just apples, the stable diffusion may generate images with different spurious correlations for the same description. Moreover, a single VLM may react differently to images (with/without) spurious correlations produced by different Image Generation Models, especially if the spurious correlations are based on visual resemblance it is important to consider images produced by different Diffusion Bias and increase detail of spurious correlations to get more accurate spurious correlations for the dataset.

2. **Not all Spurious Correlations Identified by GPT4o**: Another single source bottleneck of this approach lies in Step 2, where spurious correlations are identified only by GPT 4o which might miss important spurious correlations or specifics of spurious correlations.

3. **Unclear Human Verification Process in Step 2**: In the Step 2 of the Data Curation process, while the human verification eliminates most hallucination in spurious correlation detection, its important to clarify whether this process add additional spurious correlations/ verifies error in cases where GPT 4o hallucinates in both stages of human verification.

4. **Need for Domain Specific Spuriverse** In practical scenarios spuriverse is not general enough to be trusted in critical domains such as healthcare, where a specific version of spuriverse, replacing GPT 4o, stable diffusion with specific versions is required. The current spuriverse may only be used for validation of chatbots and other general purpose LLMs.

5. **Absence of sample Responses to prompts**: While Figure 2 does provide some sample responses GPT 4o provides, its important to release sample VLM responses along with the prompts in the supplementary material. This helps the reader ascertain the level of confidence the VLM has in its responses.

**Strengths Contributions:**

1. The need for a spurious correlation benchmark for VLMs is needed for deployment in critical applications like healthcare. The paper and dataset present one of the early works in this direction, and the data collection pipeline is able to identify spurious correlations properly although the identified spurious correlations can only be a subset and may contain non-spurious attributes.
2. The paper explores finetuning the VLMs on the spurious dataset and also on the mixed set, a step towards correcting spurious decisions of VLMs.
3. The spurious aware prompting is a notable contribution, as it offers a training-free technique to increase spurious correlation awareness. The data collection pipeline albeit the human verification can be automated, therefore, training-free spurious aware results are obtainable real-time on specific datasets like healthcare.
4. Generally the paper is well written and the dataset is accessible on huggingface and the prompts used for each step and finetuning hyperparameters are provided in the supplementary material, ensuring reproducibility.

---

> ### Author Rebuttal · Authors · 2025-07-31
>
> We thank the reviewer for their thoughtful comments. We are glad that the reviewer recognized that the benchmark is needed, the fine-tuning and prompting solutions are notable, and the paper is well-written.
>
> > Stable Diffusion Bias: The depth of the spurious labels are to be explored, VLM A might associate a red apple as a spurious correlation, VLM B with green apples, if the spurious label generated by GPT 4o is just apples, the stable diffusion may generate images with different spurious correlations for the same description. Moreover, a single VLM may react differently to images (with/without) spurious correlations produced by different Image Generation Models, especially if the spurious correlations are based on visual resemblance it is important to consider images produced by different Diffusion Bias and increase detail of spurious correlations to get more accurate spurious correlations for the dataset.
>
> We appreciate the reviewer's suggestion on spurious correlations with deeper levels of details. This is indeed an interesting future direction as different models may each have finer-grained, and differently biased, spurious correlations. In this work, our goal is to curate a set of spurious correlations that expose errors across different models. This generalizability is achieved by having human annotators verify whether the spurious correlations reflect common sense knowledge. For example, suppose tree is the target concept, while green or red apple may each be a spurious feature learned by different VLMs, an annotator will consider apple as the spurious feature.
>
> We agree that including images from different Image Generation Models would enhance the diversity of the dataset. However, at the time of the benchmark curation, most image generation models are not capable of generating images faithful to the description. We have explored different versions of Stable Diffusion (Ultra, Core, 3.5), and DALL-E3. We observed that most models (Core, 3.5, DALL-E3) hallucinate when generating the spurious or core features. We will add examples of images generated from these models and prompts used in the Appendix. We found Stable Diffusion Ultra to be the most faithful at the time. In step 4, we have annotators manually verify the generated images and edit the scene descriptions if needed.
>
> > Not all Spurious Correlations Identified by GPT4o: Another single source bottleneck of this approach lies in Step 2, where spurious correlations are identified only by GPT 4o which might miss important spurious correlations or specifics of spurious correlations.
>
> (We use the same response to reviewer DiUe here as the same question is asked.)
>
> Indeed, the early steps of our pipeline depend on GPT-4o’s errors and might introduce biases into the benchmark where there might be spurious patterns missed by GPT-4o. The biases can happen in two ways:
>
> (1) The spurious correlations found are biased to likely be model-specific if we only use the error set of GPT-4o and do human-annotation on those samples.
>
> Step 5 alleviates this bias by introducing a verification where a sample is selected when any of the VLMs (GPT-4o, Gemini 2.0 Flash, Qwen-VL-Max) has an accuracy drop between the spurious group and the core group by a threshold of 30%. This step introduces other VLMs to show that our samples are not solely curated for GPT-4o. In fact, among the 124 spurious correlations, 87 are from GPT-4o, 93 are from Gemini 2.0 Flash, and 86 are from Qwen-VL-Max, with overlaps among the models. That is, while other models might change the distribution of images, all generated images would need to expose differential behavior in some model to be included in the benchmark, and we see a balanced distribution in practice.
>
> (2) The error set to begin with is biased such that it might miss several spurious patterns appearing in other VLMs but not GPT-4o. This bias makes our artifact only a subset of the total spurious correlations in the base benchmarks we use.
>
> The goal of this paper is not to cast the most complete set to estimate all spurious correlation samples from the base benchmarks we source. Instead, we use those benchmarks as a starting point and data source to search for spurious correlations. Our proposed pipeline’s value is not to provide an estimate of spurious correlation samples in a collection of benchmarks. Instead, we use the proposed pipeline to compile a set of spurious correlation samples that are verified by humans and that VLMs struggle with. One can certainly extend our pipeline to introduce more VLMs in the early stage of the pipeline to provide a closer estimate of all spurious correlation samples from the benchmarks we source, which is not the goal of this paper.
>
> > Unclear Human Verification Process in Step 2: In the Step 2 of the Data Curation process, while the human verification eliminates most hallucination in spurious correlation detection, its important to clarify whether this process add additional spurious correlations/ verifies error in cases where GPT 4o hallucinates in both stages of human verification.
>
> In step 2, the human annotators are instructed to 1) validate whether the GPT-4o suggested errors can actually be attributed to spurious correlations 2) refine the proposed spurious features if they do not align with common sense knowledge. Humans may distill their biases during both validation and refinement. We thus include an additional verification in step 5, where multiple VLMs are used to evaluate groups with and without spurious features, and select only those where the features make a significant difference in the predictions.
>
> > Need for Domain Specific Spuriverse In practical scenarios spuriverse is not general enough to be trusted in critical domains such as healthcare, where a specific version of spuriverse, replacing GPT 4o, stable diffusion with specific versions is required. The current spuriverse may only be used for validation of chatbots and other general purpose LLMs.
>
> We appreciate the advice on domain specific Spuriverse, which is an important future direction. In this work, we aim to take an initial step to study spurious correlations for general purpose VLMs, future work can apply this process for critical domains such as medicine.
>
> > Absence of sample Responses to prompts: While Figure 2 does provide some sample responses GPT 4o provides, it is important to release sample VLM responses along with the prompts in the supplementary material. This helps the reader ascertain the level of confidence the VLM has in its responses.
>
> We thank the reviewer for the suggestion. We will include sample VLM responses along with prompts in the supplementary material.

---

> > ### Comment · Reviewer_hS21 · 2025-08-07
> > **Thanks for the rebutal**
> >
> > Thanking the authors for the rebuttal,  it has cleared my concerns, I accept that the lack of reliable diffusion models is understandable, although could have been dealt with human intervention, but given the early nature of the presented dataset, I choose to up my score to accept

---

> ### Comment · Area_Chair_GJnY · 2025-08-05
>
> Dear Reviewer hS21,
>
> Can you check the response and also read all other reviews and see if they will have any impact on your rating? Please post your first response as soon as possible within the author-reviewer discussion window (July 31 - Aug 6) so that there is time for back and forth discussion with the authors.
>
> Thanks,
> AC

---

### Official Review · Reviewer_DiUe · 2025-07-03

**Rating:** 5
**Confidence:** 3

**Summary:**

This paper introduces SpuriVerse, a novel benchmark designed to evaluate the susceptibility of Large Vision-Language Models (LVLMs) to spurious correlations in real-world VQA tasks. Rather than relying on contrived correlations from prior benchmarks, SpuriVerse extracts 124 naturally occurring spurious correlations from the failure cases of GPT-4o on multiple VQA benchmarks. Each identified spurious correlation is paired with 10 synthetic counterfactual images, producing a total of 1364 multiple-choice VQA questions. The benchmark is used to evaluate 15 LVLMs, and the authors demonstrate that all models, including state-of-the-art systems, perform poorly (worse than random guess in many cases). The paper further explores prompt-based and fine-tuning interventions, showing that while prompting is insufficient, fine-tuning on diverse spurious patterns can improve generalization to unseen spurious correlations—but with a trade-off in non-spurious performance.

**Additional Feedback:**

- It would be interesting to explore whether the benchmark design can be extended beyond VQA (e.g., captioning, retrieval).
- The multi-stage verification pipeline is robust but resource-intensive. Future work could investigate other scaling strategies for larger benchmarks.
- Is the distribution of correct answer choices uniform? Assuming each question has four choices as shown in Figure 1, random guessing should yield an accuracy of approximately 25%.

**Dataset Code Accessibility:**

Yes

**Dataset Code Comments:**

The authors release both the SpuriVerse dataset on HuggingFace and code on GitHub, enabling reproducibility. Documentation and access instructions are referenced in the Appendix and meet NeurIPS standards. While some variability in human annotation is acknowledged, the release is detailed and sufficient for research purposes.

**Ethical Comments:**

The paper takes a careful and responsible approach to evaluating model vulnerabilities. Ethical risks related to potential misuse are minimal. The work contributes positively by highlighting robustness gaps and encouraging better model training and evaluation.

**Ethical Considerations:**

No, there are no or only very minor ethics concerns

**Final Justification:**

The authors have addressed my concerns regarding the potential biases in the dataset curation process. Their response clarified the steps taken to mitigate these biases and provided additional details that were not included in the original submission. While some minor questions remain about the generalizability of the approach, I believe the main issues I raised have been adequately resolved. Overall, I find the contribution valuable, and the clarifications during the rebuttal helped strengthen the paper.

**Limitations Weaknesses:**

- **Annotation Scale and Subjectivity**: Only 194 human-verified errors (out of >11K initial errors) are retained, raising concerns about the scalability and potential biases of human annotators’ subjective judgment in identifying spurious features.
- **Statistical Significance**: Main evaluations are conducted over one seed, which limits statistical robustness. Only fine-tuning experiments include standard deviations over multiple runs.
- **Dependence on GPT-4o Errors**: The benchmark construction heavily depends on GPT-4o’s errors. While GPT-4o is strong, this introduces model bias into benchmark formation and may overlook spurious patterns missed by GPT-4o but captured by others.

**Strengths Contributions:**

- **Novel Benchmark Design**: The paper introduces a first-of-its-kind, systematically constructed and realistic benchmark for testing spurious correlation susceptibility in LVLMs using errors from GPT-4o and counterfactual generation.
- **High Quality Curation Process**: The multi-stage pipeline involving human-LVLM collaboration ensures strong signal quality and careful attribution of spurious patterns. This methodological rigor enhances the benchmark’s validity.
- **Diversity of Correlations**: SpuriVerse covers 124 distinct types of spurious correlations across six well-justified categories, such as object co-occurrence and visual predominance. This improves over existing datasets that focus on only a few contrived cases.
- **Insightful Empirical Findings**:
    - All evaluated models perform worse than random guessing on anchor samples, indicating a pervasive reliance on spurious features.
    - Prompting strategies, including CoT and “Spurious Aware,” show limited effectiveness.
    - Fine-tuning on synthetic spurious samples dramatically improves robustness to unseen spurious correlations.
    - There exists a robustness-performance trade-off between spurious and non-spurious samples, highlighting a shortcut learning phenomenon.

---

> ### Author Rebuttal · Authors · 2025-07-31
>
> We thank the reviewer for their thoughtful comments. We are glad that the reviewer recognized that the benchmark design is novel, the curation process is rigorous, the dataset is diverse, and the findings are insightful.
>
> > Annotation Scale and Subjectivity: Only 194 human-verified errors (out of >11K initial errors) are retained, raising concerns about the scalability and potential biases of human annotators’ subjective judgment in identifying spurious features.
>
> Annotation Bias – To prevent biases from human judgments, our pipeline includes an objective verification step (step 5) as a guardrail, where we evaluate multiple VLMs (GPT-4o, Gemini 2.0 Flash, Qwen-VL-Max) on the core and spurious synthetic images, and select the samples with at least a 30% accuracy gap for any of the models. This ensures that the annotated spurious features indeed contribute to the model’s error.
>
> Scalability – In our pipeline, we include human annotation efforts to ensure the quality of the spurious correlations found. However, instead of pure human annotation efforts, we use VLM to filter potential spurious samples in step 2 of the pipeline (Figure 2). This approach balances the trade-off that helps reduce the bulk of human annotation efforts (from >11K initial errors to 1.7k candidate errors) with better scalability while maintaining the quality of samples. Furthermore, our annotation process does not require domain expertise. The annotation process can be further scaled up by crowdsourcing with more base benchmarks for future work.
>
> > Statistical Significance: Main evaluations are conducted over one seed, which limits statistical robustness. Only fine-tuning experiments include standard deviations over multiple runs.
>
> We are running the main evaluations with 5 seeds and will include the results during the discussion period.
>
> > Dependence on GPT-4o Errors: The benchmark construction heavily depends on GPT-4o’s errors. While GPT-4o is strong, this introduces model bias into benchmark formation and may overlook spurious patterns missed by GPT-4o but captured by others.
>
> Indeed, the early steps of our pipeline depend on GPT-4o’s errors and might introduce biases into the benchmark where there might be spurious patterns missed by GPT-4o. The biases can happen in two ways:
>
> (1) The spurious correlations found are biased to likely be model-specific if we only use the error set of GPT-4o and do human-annotation on those samples.
>
> Step 5 alleviates this bias by introducing a verification where a sample is selected when any of the VLMs (GPT-4o, Gemini 2.0 Flash, Qwen-VL-Max) has an accuracy drop between the spurious group and the core group by a threshold of 30%. This step introduces other VLMs to show that our samples are not solely curated for GPT-4o. In fact, among the 124 spurious correlations, 87 are from GPT-4o, 93 are from Gemini 2.0 Flash, and 86 are from Qwen-VL-Max, with overlaps among the models. That is, while other models might change the distribution of images, all generated images would need to expose differential behavior in some model to be included in the benchmark, and we see a balanced distribution in practice.
>
> (2) The error set to begin with is biased such that it might miss several spurious patterns appearing in other VLMs but not GPT-4o. This bias makes our artifact only a subset of the total spurious correlations in the base benchmarks we use.
>
> The goal of this paper is not to cast the most complete set to estimate all spurious correlation samples from the base benchmarks we source. Instead, we use those benchmarks as a starting point and data source to search for spurious correlations. Our proposed pipeline’s value is not to provide an estimate of spurious correlation samples in a collection of benchmarks. Instead, we use the proposed pipeline to compile a set of spurious correlation samples that are verified by humans and that VLMs struggle with. One can certainly extend our pipeline to introduce more VLMs in the early stage of the pipeline to provide a closer estimate of all spurious correlation samples from the benchmarks we source, which is not the goal of this paper.
>
> > It would be interesting to explore whether the benchmark design can be extended beyond VQA (e.g., captioning, retrieval).
>
> > The multi-stage verification pipeline is robust but resource-intensive. Future work could investigate other scaling strategies for larger benchmarks.
>
> We appreciate the reviewer’s suggestion on the future directions of extending beyond VQA and scaling the benchmark. It would indeed be interesting to see what types of spurious correlations arise when the output format is less constrained, and discover more properties regarding spurious correlations when the dataset is larger.
>
> > Is the distribution of correct answer choices uniform? Assuming each question has four choices as shown in Figure 1, random guessing should yield an accuracy of approximately 25%.
>
> Our benchmark consists of 78 questions with 2 choices, and 46 questions with 4 choices, therefore the expected accuracy of random choice is 40.7%.

---

> > ### Comment · Reviewer_DiUe · 2025-08-02
> > **Response to Rebuttal**
> >
> > Thank you to the authors for addressing my concerns. I believe they have been adequately resolved, and I am increasing my score to 5.

---

### Note · Authors · 2025-08-15

We thank the AC and reviewers for a very helpful discussion. As acknowledged by the reviewers, our core contribution is SpuriVerse, a benchmark of 124 distinct spurious correlations curated via human–LVLM collaboration and cross-model verification. Each retained pattern exhibits a ≥30% accuracy gap between its spurious and core groups for at least one model, indicating a robust, model-relevant failure mode. We show that while state-of-the-art LVLMs are broadly susceptible to these patterns, fine-tuning on a diverse set can improve robustness to unseen spurious correlations.

We are encouraged by the consensus that all concerns raised were addressed in discussion. Some notable ones are: (i) Subjectivity/scale. We pre-filter with VLMs (11k→1.7k) and then apply human verification plus multi-model checks to retain only patterns with clear causal effect; the process requires no domain expertise and scales via crowdsourcing. (ii) GPT-4o dependence. Verification uses GPT-4o, Gemini-2.0-Flash, and Qwen-VL-Max; the 124 patterns span 87/93/86 triggers respectively (with overlaps), showing they are not curated for a single model. (iii) Statistical robustness. Running the evaluation across five seeds yields quantitatively similar results. We will report the updated results in the camera-ready version.

On acceptance, we will: (a) release the dataset including core groups, prompts, and sample VLM responses; (b) add category exemplars and examples from additional image generators (Stable Diffusion Core/3.5 and DALL-E 3); (c) update figures to high-resolution; and (d) incorporate writing and clarity edits (including the abstract revision and limitations on open-generation).

We believe the work offers a scalable, pattern-level lens on spurious correlations that complements large single-pattern datasets and provides a practical testbed for robustness methods.

---

### Decision · Program_Chairs · 2025-09-18

**Decision:**

Accept (poster)

**Comment:**

This paper initially received borderline towards positive review scores: 4, 4, 4, 4. Reviewers generally recognized the meric of this work and regard the benchmark design novel for evaluating the susceptibility of large LVLMs to spurious correlations in real-world VQA tasks, the problem well-motivated, the paper well-written, and the empirical findings insightful. Reviewers also raised some concerns, mainly about 1) the annotation scale and subjectivity, 2) stable diffusion bias; 3) relatively small number of unique samples; 4) some writing clarity suggestions.

The authors have made great efforts to address these concerns. The rebuttal was persuasive. After rebuttal, the first three  reviewers all increased the scores from 4 to 5. The final ratings unanimously recommend acceptance. The AC checked the paper, rebuttal, and review comments, and recommends accepting the paper.